# Semi-relaxed Gromov-Wasserstein divergence with applications on graphs

**Cédric Vincent-Cuaz [1], Rémi Flamary [2], Marco Corneli [1,3], Titouan Vayer [4], Nicolas Courty [5]**
Univ. Côte d'Azur, Inria, Maasai, CNRS, LJAD [1]; IP Paris, CMAP, UMR 7641 [2]; MSI [3];
Univ. Lyon, Inria, CNRS, ENS de Lyon, LIP UMR 5668 [4]; Univ. Bretagne-Suf, CNRS, IRISA [5].
`{cedric.vincent-cuaz; marco.corneli; titouan.vayer}@inria.fr`
`remi.flamary@polytechnique.edu;nicolas.courty@irisa.fr`

## Abstract

Comparing structured objects such as graphs is a fundamental operation involved in many learning tasks. To this end, the Gromov-Wasserstein (GW) distance, based on Optimal Transport (OT), has proven to be successful in handling the specific nature of the associated objects. More specifically, through the nodes connectivity relations, GW operates on graphs, seen as probability measures over specific spaces. At the core of OT is the idea of conservation of mass, which imposes a coupling between all the nodes from the two considered graphs. We argue in this paper that this property can be detrimental for tasks such as graph dictionary or partition learning, and we relax it by proposing a new semi-relaxed Gromov-Wasserstein divergence. Aside from immediate computational benefits, we discuss its properties, and show that it can lead to an efficient graph dictionary learning algorithm. We empirically demonstrate its relevance for complex tasks on graphs such as partitioning, clustering and completion.

## 1 Introduction

One of the main challenges in machine learning (ML) is to design efficient algorithms that are able to learn from structured data (Battaglia et al., 2018). Learning from datasets containing such non-vectorial objects is a difficult task that involves many areas of data analysis such as signal processing (Shuman et al., 2013), Bayesian and kernel methods on graphs (Ng et al., 2018; Kriege et al., 2020) or more recently geometric deep learning (Bronstein et al., 2017; 2021) and graph neural networks (Wu et al., 2020). In terms of applications, building algorithms that go beyond Euclidean data has led to many progresses, *e.g.* in image analysis (Harchaoui & Bach, 2007), brain connectivity (Ktena et al., 2017), social networks analysis (Yanardag & Vishwanathan, 2015) or protein structure prediction (Jumper et al., 2021).

Learning from graph data is ubiquitous in a number of ML tasks. A first one is to learn graph representations that can encode the graph structure (a.k.a. *graph representation learning*). In this domain, advances on graph neural networks led to state-of-the-art end-to-end embeddings, although requiring a sufficiently large amount of labeled data (Ying et al., 2018; Morris et al., 2019; Gao & Ji, 2019; Wu et al., 2020). Another task is to find a meaningful notion of *similarity/distance* between graphs. A way to address this problem is to leverage geometric or signal properties through the use of graph kernels (Kriege et al., 2020) or other embeddings accounting for graph isomorphisms (Zambon et al., 2020). Finally, it is often of interest either to establish meaningful structural correspondences between the nodes of different graphs, also known as *graph matching* (Zhou & De la Torre, 2012; Maron & Lipman, 2018; Bernard et al., 2018; Yan et al., 2016) or to find a representative partition of the nodes of a graph, which we refer to as *graph partitioning* (Chen et al., 2014; Nazi et al., 2019; Kawamoto et al., 2018; Bianchi et al., 2020).

**Optimal Transport for structured data.** Based on the theory of Optimal Transport (OT, Peyré & Cuturi, 2019), a novel approach to graph modeling has recently emerged from a series of works. Informally, the goal of OT is to match two probability distributions under the constraint of mass conservation and in order to minimize a given matching cost. OT originally tackles the problem

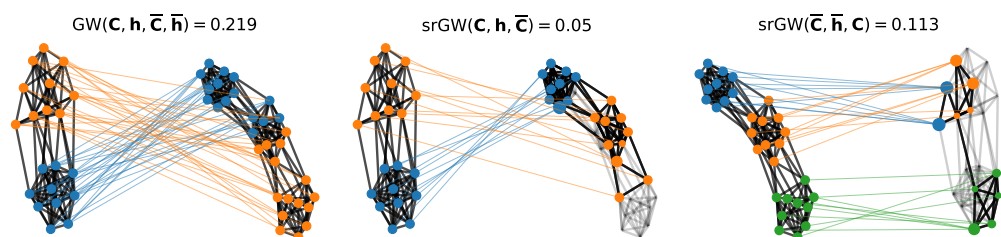

Figure 1: Comparison of the GW matching (left) and asymmetric srGW matchings (middle and right) between graphs $C$ and $\overline{C}$ with uniform distributions. Nodes of the source graph are colored based on their clusters. The OT from the source to the target nodes is represented by arcs colored depending on the corresponding source node color. The nodes in the target graph are colored by averaging the (rgb) color of the source nodes, weighted by the entries of the OT plan.

of comparing probability distributions whose supports lie on the *same* metric space, by means of the so-called Wasserstein distance. Extensions to graph data analysis were introduced by either embedding the graphs in a space endowed with Wasserstein geometry (Nikolentzos et al., 2017; Togninalli et al., 2019; Petric Maretic et al., 2019) or relying on the Gromov-Wasserstein (GW) distance (Mémoli, 2011; Sturm, 2012). The latter approach is a variant of the classical OT in which one aims at comparing probability distributions whose supports lie on *different* metric spaces by finding a matching of these distributions being as close as possible to an isometry. By expressing graphs as probability measures over spaces specific to their topology, where the structure of a graph is represented through a pairwise distance/similarity matrix between its nodes, the GW distance computes both a soft assignment matrix between the nodes of the two graphs and a notion of distance between them (see the left part of Figure 1). These properties have proven to be useful for a wide range of tasks such as graph matching and partitioning (Xu et al., 2019a; Chowdhury & Needham, 2021; Vayer et al., 2019), estimation of nonparametric graph models (*graphons*, Diaconis & Janson, 2007; Xu et al., 2021) or for graph dictionary learning (Vincent-Cuaz et al., 2021; Xu, 2020). GW was also extended to directed graphs in Chowdhury & Mémoli (2019) and to labeled graphs via the Fused Gromov-Wasserstein (FGW) distance in Vayer et al. (2020).

Despite those recent successes, applications of GW to graph modeling still have several limitations. First, finding the GW distance remains challenging, as it boils down to solving a difficult non-convex quadratic program (Peyré et al., 2016; Solomon et al., 2016; Vayer et al., 2019), which, in practice, limits the size of the graphs that can be processed. A second limit naturally emerges when a probability mass function is introduced over the set of the graph nodes. The mass associated with a node refers to its relative importance and, without prior knowledge, each node is either assumed to share the same probability mass or to have one proportional to its degree. In this paper, we somehow argue that these choices can be suboptimal in several cases, and should be relaxed, with the additional benefit of lowering the computational complexity. As an illustration, consider Figure 1: on the left image, the GW matching is given between two graphs, with respectively two and three clusters, associated with uniform weights on the nodes. By relaxing the weight constraints over the second (middle image) or first graph (right image) we obtain different matchings, that can better preserve the structure of the source graph by reweighing the target nodes and thus selecting a meaningful subgraph.

**Contributions.** We introduce a new optimal transport based divergence between graphs derived from the GW distance. We call it the **semi-relaxed Gromov-Wasserstein** (srGW) divergence. After discussing its properties and motivating its use in ML applications, we propose an efficient solver for the corresponding optimization problem. Our solver better fits to modern parallel programming than exact solvers for GW do. We empirically demonstrate the relevance of our divergence for graph partitioning, Dictionary Learning (DL), clustering of graphs and graph completion tasks. With srGW, we recover SOTA performances at a significantly lower computational cost compared to methods based on pure GW.

## 2    MODELING GRAPHS WITH THE GROMOV-WASSERSTEIN DIVERGENCE

In this section we introduce more formally the GW distance and discuss two of its applications on graphs, namely graph partitioning and unsupervised graph representation learning. In the following the probability simplex with N-bins is denoted as $\Sigma_N := \{\boldsymbol{h} \in \mathbb{R}_+^N | \sum_i h_i = 1\}$.

**GW as graphs similarity.** In the OT context, a graph $\mathcal{G}$ with $n$ nodes can be modeled as a couple $(C, h)$ where $C \in \mathbb{R}^{n \times n}$ is a matrix encoding the connectivity between nodes and $h \in \Sigma_n$ is a histogram, referred here as distribution, modeling the relative importance of the nodes within the graph. The matrix $C$ can be arbitrarily chosen as the graph adjacency matrix, or any other matrix describing the relationships between nodes in the topology of the graph (*e.g.* adjacency, shortest-path, Laplacian). The distribution is often considered as uniform ($h = \frac{1}{n}\mathbf{1}_n$) but can also convey prior knowledge *e.g.* using the normalized degree distribution (Xu et al., 2019a). Consider now two graphs $\mathcal{G} = (C, h)$ and $\overline{\mathcal{G}} = (\overline{C}, \overline{h})$, respectively with $n$ and $m$ nodes, potentially different ($n \neq m$). The GW distance between $\mathcal{G}$ and $\overline{\mathcal{G}}$ is defined as the result of the following optimization problem:

$$\mathrm{GW}_2^2(C, h, \overline{C}, \overline{h}) = \min_{\substack{T\mathbf{1}_m = h \\ T^\top \mathbf{1}_n = \overline{h}}} \sum_{ijkl} \left| C_{ij} - \overline{C}_{kl} \right|^2 T_{ik} T_{jl} \quad \text{with} \quad T \in \mathbb{R}_+^{n \times m} \tag{1}$$

The optimal coupling $T^\star$ acts as a probabilistic matching of nodes which tends to associate pairs of nodes that have similar pairwise relations in $C$ and $\overline{C}$ respectively, while preserving masses $h$ and $\overline{h}$ through its marginals constraints. GW defines a distance on the space of *metric measure* spaces (mm-spaces) invariant to measure preserving isometries (Mémoli, 2011; Sturm, 2012). In the case of graphs, which corresponds to discrete mm-spaces, such invariant is a permutation of the nodes that preserves the structures and the weights of the graphs (Vayer et al., 2020). First applications of GW to ML on graphics problems (Peyré et al., 2016; Solomon et al., 2016) motivated further connections with graph partitioning. In the GW sense, this paradigm is illustrated by finding an ideally partitioned graph $\overline{\mathcal{G}} = (\overline{D}, \overline{h})$ of $m$ clusters whose structure is a diagonal matrix $\overline{D} \in \mathbb{R}^{m \times m}$, representing the cluster's connections, and its distribution $\overline{h}$ estimates the proportion of the nodes in each cluster (Xu et al., 2019a). The OT plan between a graph $\mathcal{G} = (C, h)$ represented by its *adjacency matrix* $C$ to this ideal graph can be used to recover $\mathcal{G}$'s clusters, if their number and proportions match the number and the weights of $\overline{\mathcal{G}}$'s nodes. Xu et al. (2019a) suggested to empirically refine both graph distributions, by choosing $h$ based on power-law transformations of the degree distribution of $\mathcal{G}$ and to deduce $\overline{h}$ from $h$ by linear interpolation. Chowdhury & Needham (2021) proposed to use heat kernels from the Laplacian of $\mathcal{G}$ instead of its adjacency binary representation, and proved that the resulting GW partitioning is closely related to the well-known spectral clustering (Fiedler, 1973).

The GW distance has been extended to graphs with node attributes (typically $\mathbb{R}^d$ vectors) thanks to the Fused Gromov-Wasserstein distance (FGW) (Vayer et al., 2019). In this context, an attributed graph $\mathcal{G}$ with $n$ nodes is a tuple $\mathcal{G} = (C, F, h)$ where $F \in \mathbb{R}^{n \times d}$ is its matrix of node features. FGW between two attributed graphs aims at finding an OT by minimizing a weighted sum of a GW cost on structures and a Wasserstein cost on features (balanced by a parameter $\alpha \in [0, 1]$). Most of the applications of GW can be naturally extended with FGW to attributed graphs.

**GW for Unsupervised Graph Representation Learning.** More recently, GW has been used as a data fitting term for unsupervised graph representation learning by means of Dictionary Learning (DL) (Mairal et al., 2009; Schmitz et al., 2018). While DL methods mainly focus on vectorial data (Ng et al., 2002; Candès et al., 2011; Bobadilla et al., 2013), DL applied to graphs datasets consists in factorizing them as composition of graph primitives (or atoms) encoded as $\{(\overline{C}_k, \overline{h}_k)\}_{k \in [\![K]\!]}$. The first approach proposed by (Xu, 2020) consists in a non-linear DL based on entropic GW barycenters. On the other hand, Vincent-Cuaz et al. (2021) proposed a *linear* DL method by modeling graphs as a linear combination of graph atoms thus reducing the computational cost. In both cases, the *embedding problem* that consists in the projection of any graph on the learned graph subspace requires solving a computationally intensive optimization problem.

## 3 Semi-relaxed Gromov-Wasserstein divergence

### 3.1 Definition and properties

Given two observed graphs $\mathcal{G} = (C, h)$ and $\overline{\mathcal{G}} = (\overline{C}, \overline{h})$ of $n$ and $m$ nodes, we propose to find a correspondence between them while relaxing the weights $\overline{h}$ on the second graph. To this end we introduce the *semi-relaxed Gromov-Wasserstein divergence* as :

$$\mathrm{srGW}_2^2(C, h, \overline{C}) = \min_{\overline{h} \in \Sigma_m} \mathrm{GW}_2^2(C, h, \overline{C}, \overline{h}) \tag{2}$$

This means that we search for a reweighing of the nodes of $\overline{\mathcal{G}}$ leading to a graph with structure $\overline{C}$ with minimal GW distance from $\mathcal{G}$. While the optimization problem (2) above might seem complex to solve, it is actually equivalent to a GW problem where the mass constraints on the second marginal of $T$ are relaxed, reducing the problem to:

$$\text{srGW}_2^2(C, h, \overline{C}) = \min_{T\mathbf{1}_m = h} \sum_{ijkl} |C_{ij} - \overline{C}_{kl}|^2 T_{ik}T_{jl} \quad \text{with} \quad T \in \mathbb{R}_+^{n \times m}. \tag{3}$$

From an optimal coupling $T^\star$ of problem (3), the optimal weights $\overline{h}^\star$ expressed in problem (2) can be recovered by computing $T^\star$'s second marginal, $i.e$ $\overline{h}^\star = T^{\star\top}\mathbf{1}_n$. This reformulation with relaxed marginal has been investigated in the context of the Wasserstein distance (Rabin et al., 2014; Flamary et al., 2016) and for relaxations of the GW problem in (Schmitzer & Schnörr, 2013) but was never investigated for the GW distance itself. To the best of our knowledge, the most similar related work is the Unbalanced GW Séjourné et al. (2020); Liu et al. (2020); Chapel et al. (2019) where one could recover srGW with different weighting over the marginal relaxations ($\infty$ on the first marginal and $0$ on the second) but this specific case was not discussed nor studied in these works.

A first interesting property of srGW is that since $\overline{h}$ is optimized in the the simplex $\Sigma_m$, its optimal value $\overline{h}^\star$ can be sparse. As a consequence, parts of the graph $\overline{\mathcal{G}}$ can be omitted in the comparison, similarly to a partial matching. This behavior is illustrated in the Figure 1, where two graphs with uniform distributions and structures $C$ and $\overline{C}$ forming respectively 2 and 3 clusters are matched. The GW matching (left) between both graphs forces nodes of different clusters from $C$ to be transported on one of the three clusters of $\overline{C}$, leading to a high GW cost where clusters are not preserved. Whereas srGW can find a reasonable approximation of the structure of the left graph either though transporting on only two clusters (middle) or finding a structure with 3 clusters in a subgraph of the target graph with two clusters (right). A second natural observation resulting from the dependence of $\text{srGW}_2$ of only one input distributions is its asymmetry, $i.e.$ $\text{srGW}_2(C, h, \overline{C}) \neq \text{srGW}_2(\overline{C}, \overline{h}, C)$. Interestingly, $\text{srGW}_2$ shares similar properties than GW as described in the next proposition:

**Proposition 1** *Let $C \in \mathbb{R}^{n \times n}$ and $\overline{C} \in \mathbb{R}^{m \times m}$ be distance matrices and $h \in \Sigma_n$ with $\text{supp}(h) = [\![n]\!]$. Then $\text{srGW}_2(C, h, \overline{C}) = 0$ iff there exists $\overline{h} \in \Sigma_m$ with $\text{card}(\text{supp}(\overline{h})) = n$ and a bijection $\sigma : \text{supp}(\overline{h}) \to [\![n]\!]$ such that:*

$$\forall i \in \text{supp}(\overline{h}), \overline{h}(i) = h(\sigma(i)) \tag{4}$$

*And:*
$$\forall k, l \in \text{supp}(\overline{h})^2, \overline{C}_{kl} = C_{\sigma(k)\sigma(l)}. \tag{5}$$

In other words $\text{srGW}_2(C, h, \overline{C})$ vanishes iff there exists a reweighing $\overline{h}^\star \in \Sigma_m$ of the nodes of the second graph which cancels the GW distance. When it is the case, the induced graphs $(C, h)$ and $(\overline{C}, \overline{h}^\star)$ are isomorphic (Mémoli, 2011; Sturm, 2012). We refer the reader interested in the proofs of the equivalence and the Proposition 1 to the annex (section 7.2).

## 3.2 OPTIMIZATION AND ALGORITHMS

In this section we discuss the computational aspects of the srGW divergence and propose an algorithm to solve the related optimization problem (3). We also discuss variations resulting from entropic or/and sparse regularization of the initial quadratic problem.

**Solving for the semi-relaxed GW.** The optimization problem related to the calculation of $\text{srGW}_2^2(C, h, \overline{C})$ is a non-convex quadratic program similar to the one of GW with the important difference that the linear constraints are independent.

Consequently, we propose to solve (3) with a Conditional Gradient (CG) algorithm (Jaggi, 2013) that can benefit from those independent constraints and is known to converge to local stationary point on non-convex problems (Lacoste-Julien, 2016). This algorithm, provided in Alg.

---

**Algorithm 1** CG solver for srGW

1: **repeat**
2:    $G^{(t)} \leftarrow$ Compute gradient w.r.t $T$ of (2).
3:    $X^{(t)} \leftarrow \min_{\substack{X\mathbf{1}_m=h \\ X \geq 0}} \langle X, G^{(t)} \rangle$
4:    $T^{(t+1)} \leftarrow (1-\gamma^\star)T^{(t)} + \gamma^\star X^{(t)}$ with $\gamma^\star \in [0, 1]$ from exact-line search.
5: **until** convergence.

---

1, consists in solving at each iteration $(t)$ a linearization $\langle \boldsymbol{X}, \boldsymbol{G} \rangle$ of the problem (3) where $\boldsymbol{G}$ is the gradient of the objective in (3). The solution of the linearized problem provides a *descent direction* $\boldsymbol{X}^\star - \boldsymbol{T}$, and a linesearch whose optimal step can be found in closed form to update the current solution $\boldsymbol{T}$ (Vayer et al., 2019). While both srGW and GW CG require at each iteration the computation of a gradient with complexity $O(n^2 m + m^2 n)$, the main source of efficiency of our algorithm comes from the computation of the descent directions. In the GW case, one needs to solve an exact linear OT problem, while in our case, one just needs to solve several independent linear problems under a simplex constraint. This simply amounts to finding the minimum on the rows of $\boldsymbol{G}$ as discussed in (Flamary et al., 2016, Equation (8)), within $O(mn)$ operations, with potential parallelization with GPUs. Performance gains are illustrated in the experimental section.

**Entropic regularization.** Recent OT applications have shown the interest of adding an entropic regularization to the exact problem (3) ,*e.g.* (Cuturi, 2013; Peyré et al., 2016). Entropic regularization makes the optimization problem smooth and more robust while densifying the optimal transport plan. Similar to (Peyré et al., 2016; Xu et al., 2019b; Xie et al., 2020), we can use a *mirror-descent scheme w.r.t.* the Kullback-Leibler divergence (KL) to solve entropic regularized srGW. The problem boils down to find, at iteration $t$ the coupling $\boldsymbol{T}^{(t+1)} \leftarrow \arg\min_{\boldsymbol{T}\mathbf{1}_m = \boldsymbol{h}; \boldsymbol{T} \geq 0} \langle \boldsymbol{T}, \boldsymbol{G}^{(t)} \rangle + \epsilon \, \mathrm{KL}(\boldsymbol{T}|\boldsymbol{T}^{(t)})$ where $\epsilon > 0$, $\boldsymbol{G}^{(t)}$ denotes the gradient of the GW loss at iteration $t$ and $\mathrm{KL}(\boldsymbol{T}|\boldsymbol{T}^{(t)})$ is the KL divergence. These updates can be solved using the following closed-form Bregman Projections :

$$\boldsymbol{T}^{(t+1)} \leftarrow \underset{\boldsymbol{T}\mathbf{1}_m = \boldsymbol{h}; \boldsymbol{T} \geq 0}{\arg\min} \; \epsilon \, \mathrm{KL}(\boldsymbol{T}|\boldsymbol{K}^{(t+1)}) \quad \Leftrightarrow \quad \boldsymbol{T}^{(t+1)} \leftarrow \mathrm{diag}\left(\frac{\boldsymbol{h}}{\boldsymbol{K}^{(t)}\mathbf{1}_m}\right)\boldsymbol{K}^{(t)}. \tag{6}$$

where $\boldsymbol{K}^{(t)} = \exp\left(\boldsymbol{G}^{(t)} - \epsilon \log(\boldsymbol{T}^{(t)})\right)$ (exp and log are applied componentwise). Unlike existing solvers for GW based on entropic regularization (Peyré et al., 2016; Solomon et al., 2016) that rely on a Sinkhorn's matrix scaling algorithm on $\boldsymbol{K}^{(t)}$ *at each iteration*, our problem requires only one (left) scaling of $\boldsymbol{K}^{(t)}$ per iteration. We denote by $\mathrm{srGW}_e(\boldsymbol{C}, \boldsymbol{h}, \overline{\boldsymbol{C}})$ the result of this procedure.

**Sparsity promoting regularization.** As illustrated in Figure 1, srGW naturally leads to sparse solutions in $\bar{\boldsymbol{h}}$. To compress even more the localization over a few nodes of $\overline{\boldsymbol{C}}$, we can promote the sparsity of $\overline{\boldsymbol{h}}$ through a penalization $\Omega(\boldsymbol{T}) = \sum_j |\bar{h}_j|^{1/2}$ which defines a concave function in the positive orthant $\mathbb{R}_+$. We adapt the Majorisation-Minimisation (MM) of Courty et al. (2014) that was introduced to solve classical OT with a similar regularizer. The resulting algorithm, which relies on a local linearization of $\Omega(\boldsymbol{T}^{(t)})$, consists in iteratively solving the srGW or $\mathrm{srGW}_e$ problems, regularized at iteration $t+1$ by a linear OT cost of components $R_{i,j}^{(t)} = \frac{\lambda_g}{2}(\bar{h}_j^{(t)})^{-1/2}$. Further detailed explanations on these algorithms can be found in section 7.3 of the annex.

## 4 LEARNING THE TARGET STRUCTURE

A dataset of $K$ graphs $\mathcal{D} = \{(\boldsymbol{C}_k, \boldsymbol{h}_k)\}_{k \in [\![K]\!]}$ is now considered, with heterogeneous structures and a variable number of nodes, denoted by $(n_k)_{k \in [\![K]\!]}$. In the following, we introduce a novel graph dictionary learning (DL) whose peculiarity is to learn *a unique* dictionary element. Then we discuss how this dictionary can be used to perform graph completion, *i.e.* reconstruct the full structure of a graph from an observed subgraph.

### 4.1 A NEW GRAPH DICTIONARY LEARNING

**Semi-relaxed Gromov-Wasserstein embedding.** We first discuss how an observed graph can be represented in a dictionary with a unique element (or *atom*) $\overline{\boldsymbol{C}} \in \mathbb{R}^{m \times m}$, assumed to be known or designed through expert knowledge. First, one computes $\mathrm{srGW}_2^2(\boldsymbol{C}_k, \boldsymbol{h}_k, \overline{\boldsymbol{C}})$ using the algorithmic solutions and regularization strategies detailed in Section 3.1. From the resulting optimal coupling $\boldsymbol{T}_k^\star$, the optimal weights for the target graph $\overline{\boldsymbol{C}}$ are recovered with $\overline{\boldsymbol{h}}_k^\star = \boldsymbol{T}_k^{\star \top} \mathbf{1}_{n_k}$. The graph $(\overline{\boldsymbol{C}}, \overline{\boldsymbol{h}}_k^\star)$ can be seen as a projection of $(\boldsymbol{C}_k, \boldsymbol{h}_k)$ in the GW sense and the distribution on the nodes $\overline{\boldsymbol{h}}_k^\star$ is an embedding of the graph $(\boldsymbol{C}_k, \boldsymbol{h}_k)$. Representing a graph as a vector of weights $\overline{\boldsymbol{h}}_k^\star$ on a graph $\overline{\boldsymbol{C}}$ is a new and elegant way to define a graph subspace that is orthogonal to other DL methods that either rely on GW barycenters (Xu, 2020) or linear representations (Vincent-Cuaz et al., 2021). One particularly

---

**Algorithm 2** Stochastic update of the dictionary atom $\overline{C}$

1: Sample a minibatch of graphs $\mathcal{B} := \{(C^{(k)}, h^{(k)})\}_k$.
2: Get transports $\{T_k^\star\}_{k \in \mathcal{B}}$ from srGW$(C_k, h_k, \overline{C})$ with Alg.1.
3: Compute the gradient $\widetilde{\nabla}_{\overline{C}}$ of srGW with fixed $\{T_k^\star\}_{k \in \mathcal{B}}$ and perform a projected gradient step on symmetric non-negative matrices $\mathcal{S}$:
$$\overline{C} \leftarrow \mathrm{Proj}_{\mathcal{S}}(\overline{C} - \eta \widetilde{\nabla}_{\overline{C}}) \tag{7}$$

---

interesting aspect of this modeling is that when $\overline{h}_k^\star$ is sparse, only the subpart (or subgraph) of $\overline{C}$ corresponding to the nodes with non-zero weights in $\overline{h}_k^\star$ is used.

**srGW Dictionary Learning and online algorithm.** Given a dataset of graphs $\mathcal{D} = \{(C_k, h_k)\}_{k \in [K]}$, we propose to learn the graph dictionary $\overline{C} \in \mathbb{R}^{m \times m}$ from the observed data, by optimizing:

$$\min_{\overline{C} \in \mathbb{R}^{m \times m}} \frac{1}{K} \sum_{k=1}^{K} \mathrm{srGW}_2^2(C_k, h_k, \overline{C}). \tag{8}$$

This problem is denoted as **srGW Dictionary Learning**. It can be seen as a srGW barycenter problem (Peyré et al., 2016) where we look for a graph structure $\overline{C}$ for which there exists node weights $(\overline{h}_k^\star)_{k \in [\![K]\!]}$ leading to a minimal GW error. Interestingly this DL model requires only to solve the srGW problem to compute the embedding $\overline{h}_k^\star$ since it can be recovered from the solution $\overline{T}_k^\star$ of the problem.

We solve this non-convex optimization problem above with an online algorithm similar to the one first proposed in Mairal et al. (2009) for vectorial data and adapted by Vincent-Cuaz et al. (2021) for graph data. The core of the stochastic algorithm is depicted in Alg. 2. The main idea is to use batches of graphs to independently solve the embedding problems (via Alg.1), then compute estimates of the gradient $\widetilde{\nabla}_{\overline{C}}$ with respect to $\overline{C}$ on each batch $\mathcal{B}$. Finally a projected gradient on the set $\mathcal{S}$ of symmetric non-negative matrices is performed to update $\overline{C}$. In practice we use Adam optimizer (Kingma & Ba, 2015) in all our experiments. The complexity of this stochastic algorithm is mostly bounded by computing the gradients, which can be done in $O(n_k^2 m + m^2 n_k)$ (see Section 3.2). Hence, in a factorization context *i.e.* $\max_k (n_k) >> m$, the overall learning procedure has a quadratic complexity *w.r.t.* the maximum graphs size. Since the embedding $\overline{h}_k^\star$ is a by-product of computing the different srGW, we do not need an iterative solver to estimate it. Consequently, it leads to a speed up on CPU of 2 to 3 orders of magnitude compared to our main competitors (see Section 5.3) whose DL methods, instead, require such iterative scheme.

## 4.2 DL-BASED MODEL FOR GRAPHS COMPLETION

The structure $\overline{C}$ estimated on the dataset $\mathcal{D}$ can be used to infer/complete a new graph from the dataset that is only partially observed. In this setting, we aim at recovering the full structure $C \in \mathbb{R}^{n \times n}$ while only a subset of relations between $n_{obs} < n$ nodes is observed, denoted as $C_{obs} \in \mathbb{R}^{n_{obs} \times n_{obs}}$. This amounts to solving:

$$\min_{C_{imp}} \quad \mathrm{srGW}_2^2\left(\widetilde{C}, h, \overline{C}\right), \text{ where } \widetilde{C} = \left[ \begin{array}{c|c} C_{obs} & \vdots \\ \hline \cdots & C_{imp} \end{array} \right], \tag{9}$$

where only the $n^2 - n_{obs}^2$ coefficients collected into $C_{imp}$ are optimized (and thus imputed). We solve the optimization problem above by a classical projected gradient descent. At each iteration we find an optimal coupling $T^\star$ of srGW that is used to calculate the gradient of srGW *w.r.t* $\overline{C}_{imp}$. The latter is obtained as the gradient of the srGW cost function evaluated at the fixed optimal coupling $T^\star$ by using the Envelope Theorem (Bonnans & Shapiro, 2000). The projection step is here to enforce known properties of $C$, such as positivity and symmetry. In practice the estimated $C_{imp}$ will have continuous values, so one has to apply a thresholding (with value $0.5$) on $C_{imp}$ to recover a binary adjacency matrix. The method can be easily extended to labeled graphs by also optimizing the node features of non-observed nodes.

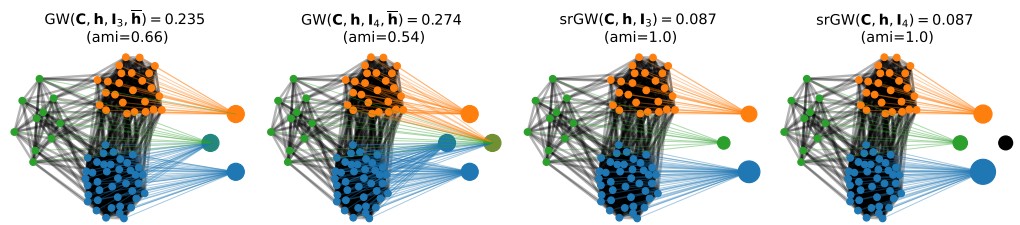

Figure 2: GW vs srGW partitioning of $\mathcal{G} = (\boldsymbol{C}, \boldsymbol{h} = \mathbf{1}_n/n)$ with 3 clusters of varying proportions to $\overline{\mathcal{G}} = (\boldsymbol{I}_q, \overline{\boldsymbol{h}})$ where $\overline{\boldsymbol{h}}$ is fixed to uniform for GW (left) and estimated for for srGW (right) for $q = 3$ and $q = 4$. Nodes of $\mathcal{G}$ are colored based on their cluster assignments while those of $\overline{\mathcal{G}}$ are interpolated based on linear interpolation of node colors of $\mathcal{G}$ linked to them through their OT (colored line between nodes) if these links exist, otherwise default nodes color is black. Node sizes of both graphs $\mathcal{G}$ and $\overline{\mathcal{G}}$ depend on their respective masses $\boldsymbol{h}$ and $\overline{\boldsymbol{h}}$.

## 5 NUMERICAL EXPERIMENTS

This section illustrates the behavior of srGW on graph partitioning (a.k.a. nodes clustering), clustering of graphs and graph completion. All the Python implementations in the experiments will be released on Github. For all experiments we provide *e.g.* validation grids, initializations and complementary metrics in the annex (see sections: partitioning 7.5.1, clustering 7.5.2, completion 7.5.3).[1]

### 5.1 GRAPH PARTITIONING

As discussed in Section 2, it is possible to achieve graph partitioning via OT by estimating a GW matching between the graph to partition $\mathcal{G} = (\boldsymbol{C}, \boldsymbol{h})$ ($n$ nodes) and a smaller graph $\overline{\mathcal{G}} = (\boldsymbol{I}_q, \overline{\boldsymbol{h}})$, with $q \ll n$ nodes. The atom $\boldsymbol{I}_q$ is set as the identity matrix to enforce the emergence of densely connected groups (i.e. *communities*). The distribution $\overline{\boldsymbol{h}}$ estimates the proportion of nodes in each cluster. We recall that $\overline{\boldsymbol{h}}$ must be given to compute the GW distance, whereas it is estimated with srGW. All partitioning performances are measured by Adjusted Mutual Information (AMI, Vinh et al., 2010).

**Simulated data.** In Figure 2, we illustrate the behavior of GW and srGW partitioning on a toy graph, simulated according to SBM (Holland et al., 1983) with 3 clusters of different proportions. We see that miss-classification occurs either when the proportions $\overline{\boldsymbol{h}}$ do not fit the true ones or when $q \neq 3$. On the other hand, clustering from srGW can simultaneously recover any cluster proportions (since it estimates them) and can select the actual number of clusters, using the sparse properties of $\overline{\boldsymbol{h}}$.

**Real-world datasets.** In order to benchmark the srGW partitioning on real (directed and undirected) graphs, we consider 4 datasets (details provided in Table 4 of the annex): a Wikipedia hyperlink network (Yang & Leskovec, 2015); a directed graph of email interactions between departments of a European research institute (Yin et al., 2017); an Amazon product network (Yang & Leskovec, 2015); a network of interactions between indian villages (Banerjee et al., 2013). For the directed graphs, we adopt the symmetrization

Table 1: Partitioning performances on real datasets measured by AMI. We see in bold (resp. italic) the first (resp. second) best method. NA: non applicable.

|  | Wikipedia | | EU-email | | Amazon | Village |
| --- | --- | --- | --- | --- | --- | --- |
|  | asym | sym | asym | sym | sym | sym |
| srGW (ours) | 56.92 | 56.92 | 49.94 | 50.11 | 48.28 | 81.84 |
| srSpecGW | 50.74 | **63.07** | 49.08 | 50.60 | 76.26 | *87.53* |
| srGW$_e$ | **57.13** | *57.55* | **54.75** | *55.05* | 50.00 | 83.18 |
| srSpecGW$_e$ | 53.76 | 61.38 | *54.27* | 50.89 | *85.10* | 84.31 |
| GWL | 38.67 | 35.77 | 47.23 | 46.39 | 38.56 | 68.97 |
| SpecGWL | 40.73 | 48.98 | 45.89 | 49.02 | 65.16 | 77.85 |
| FastGreedy | NA | 55.30 | NA | 45.89 | 77.21 | **93.66** |
| Louvain | NA | 54.72 | NA | **56.12** | 76.30 | **93.66** |
| InfoMap | 46.43 | 46.43 | 54.18 | 49.10 | **94.33** | **93.66** |

procedure described in Chowdhury & Needham (2021). Our main competitors are the two GW based partitioning methods proposed by Xu et al. (2019b) and Chowdhury & Needham (2021). The former (GWL) relies on adjacency matrices, the latter (SpecGWL) adopts heat kernels on the graph normalized laplacians (SpecGWL). The GW solver of Flamary et al. (2021) was used for these methods. For fairness, we also consider these two representations for srGW partitioning (namely srGW and srSpecGW). Finally, three competing methods specialized in graph partitioning are also considered: FastGreedy (Clauset et al., 2004), Louvain (with validation of its resolution parameter,

---

[1]code available at `https://github.com/cedricvincentcuaz/srGW`.

Table 2: Clustering performances on real datasets measured by Rand Index. In bold (resp. italic) we highlight the first (resp. second) best method.

| MODELS | NO ATTRIBUTE | | DISCRETE ATTRIBUTES | | REAL ATTRIBUTES | | | |
|---|---|---|---|---|---|---|---|---|
| | IMDB-B | IMDB-M | MUTAG | PTC-MR | BZR | COX2 | ENZYMES | PROTEIN |
| srGW (ours) | 51.59(0.10) | 54.94(0.29) | 71.25(0.39) | 51.48(0.12) | 62.60(0.96) | 60.12(0.27) | 71.68(0.12) | 59.66(0.10) |
| srGW$_g$ | **52.44(0.46)** | **56.70(0.34)** | 72.31(0.51) | 51.76(0.39) | 66.75(0.38) | **62.15(0.27)** | **72.51(0.10)** | *60.67(0.29)* |
| srGW$_e$ | 51.75(0.56) | 55.36(0.14) | *74.41(0.84)* | *52.35(0.42)* | *67.63(1.17)* | 59.75(0.39) | 70.93(0.33) | 59.97(0.21) |
| srGW$_{e+g}$ | *52.23(0.83)* | *55.90(0.68)* | **74.69(0.73)** | **52.53(0.47)** | **67.81(0.94)** | *60.53(0.36)* | 71.31(0.52) | **60.81(0.43)** |
| GDL | 51.34(0.27) | 55.14(0.35) | 70.28(0.25) | 51.49(0.31) | 62.84(1.60) | 58.39(0.52) | 69.83(0.33) | 60.19(0.28) |
| GDL$_{reg}$ | 51.69(0.56) | 55.43(0.22) | 70.92(0.11) | 51.82(0.47) | 66.30(1.71) | 59.61(0.74) | 71.03(0.36) | 60.46(0.65) |
| GWF-r | 51.02(0.30) | 55.09(0.46) | 69.07(1.02) | 51.47(0.59) | 52.45(2.41) | 56.91(0.46) | *72.09(0.21)* | 59.97(0.11) |
| GWF-f | 50.43(0.29) | 54.18(0.27) | 59.13(1.87) | 50.82(0.81) | 51.75(2.84) | 52.84(0.53) | 71.58(0.31) | 58.92(0.41) |
| GW-k | 50.31(0.03) | 53.67(0.07) | 57.62(1.45) | 50.42(0.33) | 56.77(0.53) | 52.45(0.13) | 66.35(1.37) | 50.08(0.01) |

Blondel et al., 2008) and Infomap (Rosvall & Bergstrom, 2008). The graph partitioning results are reported in Table 1. Our method srGW always outperforms the GW based approaches and on this application the entropic regularization seems to improve the performance. We want to stress that our *general purpose* divergence srGW outperforms methods that have been specifically designed for nodes clustering tasks on 3 out of 6 datasets.

## 5.2 CLUSTERING OF GRAPHS DATASETS

**Datasets and methods.** We now show how the embeddings $(\overline{h}_k^\star)_{k \in [\![K]\!]}$ provided by the srGW Dictionary Learning can be particularly useful for the task of graphs clustering. We considered here three types of datasets (details provided in Table 9 of the annex): i) social networks from IMDB-B and IMDB-M (Yanardag & Vishwanathan, 2015); ii) graphs with discrete features representing chemical compounds from MUTAG (Debnath et al., 1991) and cuneiform signs from PTC-MR (Krichene et al., 2015); iii) graphs with continuous features, namely BZR, COX2 (Sutherland et al., 2003) and PROTEINS, ENZYMES (Borgwardt & Kriegel, 2005). Our main competitors are the following OT-based SOTA models: i) GDL (Vincent-Cuaz et al., 2021) and its regularized version, namely GDL$_\lambda$; ii) GWF (Xu, 2020), with both fixed (GWF-f) and random (GWF-r, default setting for the method) atom size; iii) GW kmeans (GW-k), a k-means equipped with GW distances and barycenters (Peyré et al., 2016).

**Experimental settings.** For all experiments we follow the benchmark proposed in Vincent-Cuaz et al. (2021). The clustering performances are measured by means of Rand Index (RI, Rand, 1971). The standard Euclidean distance is used to implement k-means over srGW and GWFs embeddings, but we use for GDL the dedicated Mahalanobis distance as described in Vincent-Cuaz et al. (2021). GW-k does not use any embedding since it directly computes (a GW) k-means over the input graphs. For each parameter configuration (number of atoms, number of nodes and regularization parameter, detailed in section 7.5.2) we run each experiment five times, independently. The mean RI over the five runs is computed and the dictionary configuration leading to the highest RI for each method is reported.

**Results and discussion.** Clustering performances and running times are reported in Tables 2 and 3, respectively. All variants of srGW DL are at least comparable with the SOTA GW based methods. Remarkably, the sparsity promoting variants always outperform other methods. Notably Table 3 shows embedding computation times of the order of the millisecond for srGW, two order of magnitude faster than the competitors.

## 5.3 GRAPHS COMPLETION

Finally, we present graph completion results on the real world datasets IMDB-B and MUTAG, using the approach proposed in 4.2. Since this completion problem has never been investigated by existing GW graph methods, we adapted the learning procedure used for srGW to GDL (Vincent-Cuaz et al., 2021).

**Experimental setting.** Since the datasets do not explicitly contain graphs with missing nodes, we proceed as follow: first we split the dataset into a training dataset ($\mathcal{D}_{train}$) used to learn the dictionary and a test dataset ($\mathcal{D}_{test}$) reserved for the completion tasks. For each graph of $C \in \mathcal{D}_{test}$, we created incomplete graphs $C_{obs}$ by independently removing $10\%$ and $20\%$ of their nodes, uniformly

Table 3: Embedding computation times (in ms) averaged over whole datasets at a convergence precision of $10^{-5}$ on learned dictionaries. $(-)$ (resp. $(+)$) denotes the fastest (resp. slowest) runtimes regarding DL configurations. We report here runtimes using $FGW_{0.5}$ for datasets with nodes attributes. Measures taken on Intel(R) Core(TM) i7-4510U CPU @ 2.00GHz.

| | NO ATTRIBUTE | | | | DISCRETE ATTRIBUTES | | | | REAL ATTRIBUTES | | | | | | | |
|---|---|---|---|---|---|---|---|---|---|---|---|---|---|---|---|---|
| | IMDB-B | | IMDB-M | | MUTAG | | PTC-MR | | BZR | | COX2 | | ENZYMES | | PROTEIN | |
| | (-) | (+) | (-) | (+) | (-) | (+) | (-) | (+) | (-) | (+) | (-) | (+) | (-) | (+) | (-) | (+) |
| srGW (ours) | 1.51 | 2.62 | 0.83 | 1.59 | 0.86 | 1.83 | 0.40 | 1.01 | 0.43 | 0.79 | 0.51 | 0.90 | 0.62 | 0.95 | 0.46 | 0.60 |
| srGW$_g$ | 1.95 | 6.11 | 1.06 | 5.53 | 3.68 | 5.98 | 1.65 | 3.38 | 0.89 | 2.88 | 0.97 | 4.60 | 1.35 | 4.73 | 1.57 | 2.96 |
| GWF-f | 219 | 651 | 103 | 373 | 236 | 495 | 191 | 477 | 181 | 916 | 129 | 641 | 93 | 627 | 78 | 322 |
| GDL | 108 | 236 | 43.8 | 152 | 102 | 514 | 100 | 509 | 73.2 | 532 | 48.7 | 347 | 38 | 301 | 29 | 151 |

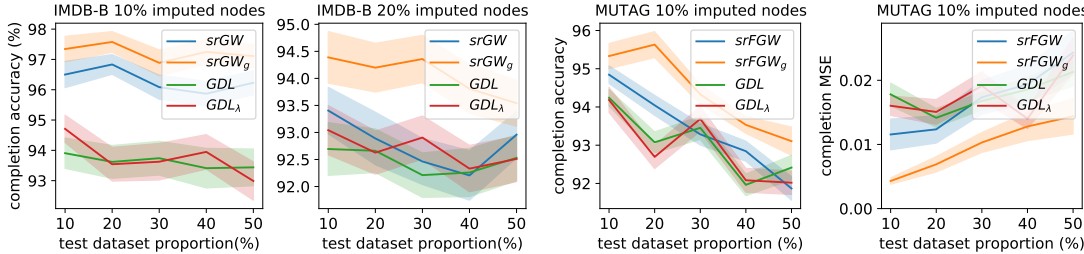

Figure 3: Completion performances for IMDB-B (left) and MUTAG (right) datasets, measured by means of accuracy for structures and Mean Squared Error for node features, respectively averaged over all imputed graphs.

at random. The partially observed graphs are then reconstructed using the procedure described in Section 4.2 and the average performance of each method is computed for 5 different dataset splits.

**Results.** The graph completion results are reported in Figure 3. Our srGW dictionary learning outperforms GDL consistently, when enough data is available to learn the atoms. When the proportion of train/test data varies, we can see that the linear GDL model that maintains the marginal constraints tends to be less sensitive to the scarcity of data. This can come form the fact that srGW is more flexible thanks to the optimization of $\overline{h}$ but can slightly overfit when few data is available. Sparsity promoting regularization can clearly compensate this overfitting and systematically leads to the best completion performances (high accuracy, low Means Square Error).

# 6 CONCLUSION

We introduce a new transport based divergence between structured data by relaxing the mass constraint on the second distribution of the Gromov-Wasserstein problem. After designing efficient solvers to estimate this divergence, called the semi-relaxed Gromov-Wasserstein (srGW), we suggest to learn a unique structure to describe a dataset of graphs in the srGW sense. This novel modeling can be seen as a Dictionary Learning approach where graphs are embedded as a subgraph of a single atom. Numerical experiments highlight the interest of our methods for graph partitioning, and unsupervised representation learning whose evaluation is conducted through clustering and completion of graphs.

We believe that this new divergence will unlock the potential of GW for graphs with unbalanced proportions of nodes. The associated fast numerical solvers allow to handle large size graph datasets, which was not possible with current GW solvers. One interesting future research direction includes an analysis of srGW to perform parameters estimation of stochastic block models. Also, as relaxing the second marginal constraint in the original optimization problem gives more degrees of freedom to the underlying problem, one can expect dedicated regularization schemes, over *e.g.* the level of sparsity of $\overline{h}$, to address a variety of application needs. Finally, our method can be seen as a special relaxation of the subgraph isomorphism problem. It remains to be understood theoretically in which sense this relaxation holds.

## ACKNOWLEDGMENTS

This work is partially funded through the projects OATMIL ANR-17-CE23-0012, OTTOPIA ANR-20-CHIA-0030 and 3IA Côte d'Azur Investments ANR-19-P3IA-0002 of the French National Research Agency (ANR). This research was produced within the framework of Energy4Climate Interdisciplinary Center (E4C) of IP Paris and Ecole des Ponts ParisTech. This research was supported by 3rd Programme d'Investissements d'Avenir ANR-18-EUR-0006-02. This action benefited from the support of the Chair "Challenging Technology for Responsible Energy" led by l'X Ecole polytechnique and the Fondation de l'Ecole polytechnique, sponsored by TOTAL. This work is supported by the ACADEMICS grant of the IDEXLYON, project of the Universit de Lyon, PIA operated by ANR-16-IDEX-0005. The authors are grateful to the OPAL infrastructure from Université Côte d'Azur for providing resources and support.

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

# 7 APPENDIX

## 7.1 NOTATIONS & DEFINITIONS

In this section we recall the notations used in the rest of the appendix.

**Notations.** For a vector $\boldsymbol{h} \in \mathbb{R}^m$ we define its support as $\mathrm{supp}(\boldsymbol{h}) = \{i \in [\![m]\!] | h_i \neq 0\}$. Note that if $\boldsymbol{h} \in \Sigma_m$ we have $\mathrm{supp}(\boldsymbol{h}) = \{i \in [\![m]\!] | h_i > 0\}$. The cardinal of a discrete set $A$ is denoted $|A|$.

For a 4-D tensor $\boldsymbol{L} = (L_{ijkl})_{ijkl}$ we denote $\otimes$ the tensor-matrix multiplication such that for a given matrix $\boldsymbol{M}$, $\boldsymbol{L} \otimes \boldsymbol{M}$ is the matrix with entries $\left(\sum_{kl} L_{ijkl} M_{kl}\right)_{ij}$.

**Constraints on couplings.** We introduce $\mathcal{U}(\boldsymbol{h}, \overline{\boldsymbol{h}})$ the set of all admissible couplings between $\boldsymbol{h}$ and $\overline{\boldsymbol{h}}$, i.e the set

$$\mathcal{U}(\boldsymbol{h}, \overline{\boldsymbol{h}}) := \{\boldsymbol{T} \in \mathbb{R}_+^{n \times m} | \boldsymbol{T} \mathbf{1}_m = \boldsymbol{h}, \boldsymbol{T}^\top \mathbf{1}_n = \overline{\boldsymbol{h}}\}.$$

We also introduce for any $\boldsymbol{h} \in \Sigma_n$ and $m \in \mathbf{N}^*$, $\mathcal{U}(\boldsymbol{h}, m)$ the set of all admissible couplings between $\boldsymbol{h}$ and any histogram of $\Sigma_m$, i.e the set

$$\mathcal{U}(\boldsymbol{h}, m) := \{\boldsymbol{T} \in \mathbb{R}_+^{n \times m} | \boldsymbol{T} \mathbf{1}_m = \boldsymbol{h}\},$$

such that $\forall \boldsymbol{T} \in \mathcal{U}(\boldsymbol{h}, m)$, the second marginal $\overline{\boldsymbol{h}}(= \boldsymbol{T}^\top \mathbf{1}_n)$ belongs to $\Sigma_m$.

**Gromov-Wasserstein distance.** For any $q \geq 1$, the Gromov-Wasserstein distance of order $q$ between $(\boldsymbol{C}, \boldsymbol{h})$ and $(\overline{\boldsymbol{C}}, \overline{\boldsymbol{h}})$ is defined by:

$$\mathrm{GW}_q^q(\boldsymbol{C}, \boldsymbol{h}, \overline{\boldsymbol{C}}, \overline{\boldsymbol{h}}) = \min_{\boldsymbol{T} \in \mathcal{U}(\boldsymbol{h}, \overline{\boldsymbol{h}})} \mathcal{L}_q(\boldsymbol{C}, \overline{\boldsymbol{C}}, \boldsymbol{T})$$

where

$$\mathcal{L}_q(\boldsymbol{C}, \overline{\boldsymbol{C}}, \boldsymbol{T}) = \sum_{i,j \in [\![n]\!]^2} \sum_{k,l \in [\![m]\!]^2} |C_{ij} - \overline{C}_{kl}|^q T_{ik} T_{jl}.$$

This definition can be equivalently expressed with a tensor-matrix multiplication as

$$GW_q^q(\boldsymbol{C}, \boldsymbol{h}, \overline{\boldsymbol{C}}, \overline{\boldsymbol{h}}) = \min_{\boldsymbol{T} \in \mathcal{U}(\boldsymbol{h}, \overline{\boldsymbol{h}})} \langle \boldsymbol{L}_q(\boldsymbol{C}, \overline{\boldsymbol{C}}) \otimes \boldsymbol{T}, \boldsymbol{T} \rangle \tag{10}$$

where $\boldsymbol{L}_q(\boldsymbol{C}, \overline{\boldsymbol{C}})$ is the 4-D tensor such that $\boldsymbol{L}_q(\boldsymbol{C}, \overline{\boldsymbol{C}}) = \big((C_{ij} - \overline{C}_{kl})^q\big)_{ijkl}$.

**Semi-relaxed Gromov-Wasserstein divergence.** The semi-relaxed Gromov-Wasserstein divergence of order $q$ satisfies the following equation:

$$\mathrm{srGW}_q^q(\boldsymbol{C}, \boldsymbol{h}, \overline{\boldsymbol{C}}) = \min_{\boldsymbol{T} \in \mathcal{U}(\boldsymbol{h}, m)} \mathcal{L}_q(\boldsymbol{C}, \overline{\boldsymbol{C}}, \boldsymbol{T})$$

or equivalently

$$\mathrm{srGW}_q^q(\boldsymbol{C}, \boldsymbol{h}, \overline{\boldsymbol{C}}) = \min_{\boldsymbol{T} \in \mathcal{U}(\boldsymbol{h}, m)} \langle \boldsymbol{L}_q(\boldsymbol{C}, \overline{\boldsymbol{C}}) \otimes \boldsymbol{T}, \boldsymbol{T} \rangle. \tag{11}$$

**Fused Gromov-Wasserstein distance.** For attributed graphs i.e graphs with nodes features, we use the Fused Gromov-Wasserstein distance (Vayer et al., 2019). Consider two attributed graphs $\mathcal{G} = (\boldsymbol{C}, \boldsymbol{F}, \boldsymbol{h})$ and $\overline{\mathcal{G}} = (\overline{\boldsymbol{C}}, \overline{\boldsymbol{F}}, \overline{\boldsymbol{h}})$ where $\boldsymbol{F} = (F_i)_{i \in [\![n]\!]} \in \mathbb{R}^{n \times d}$ and $\overline{\boldsymbol{F}} = (\overline{F}_j)_{j \in [\![m]\!]} \in \mathbb{R}^{m \times d}$ are their respective matrix of features whose rows are denoted by $\{\boldsymbol{F}_i\}_{i \in [\![n]\!]}$ and $\{\overline{\boldsymbol{F}}_i\}_{i \in [\![m]\!]}$. Given a trade-off parameter between structures and features denoted $\alpha \in [0; 1]$, the $FGW_{q,\alpha}^q$ distance is defined as the result of the following optimization problem

$$\min_{\boldsymbol{T} \in \mathcal{U}(\boldsymbol{h}, \overline{\boldsymbol{h}})} (1 - \alpha) \sum_{ij} \|\mathbf{F_i} - \overline{\mathbf{F}_j}\|_q^q T_{ij} + \alpha \sum_{ijkl} |C_{ij} - \overline{C}_{kl}|^q T_{ik} T_{jl} \tag{12}$$

which is equivalent to

$$\min_{\boldsymbol{T} \in \mathcal{U}(\boldsymbol{h}, \overline{\boldsymbol{h}})} \langle (1 - \alpha) \boldsymbol{M}_q(\boldsymbol{F}, \overline{\boldsymbol{F}}) + \alpha \boldsymbol{L}_q(\boldsymbol{C}, \overline{\boldsymbol{C}}) \otimes \boldsymbol{T}, \boldsymbol{T} \rangle \tag{13}$$

where $\boldsymbol{M}_q(\boldsymbol{F}, \overline{\boldsymbol{F}})$ is a matrix with entries $\boldsymbol{M}_q(\boldsymbol{F}, \overline{\boldsymbol{F}})_{ij} = \|\mathbf{F_i} - \overline{\mathbf{F}_j}\|_q^q$.

**Semi-relaxed Gromov-Wasserstein distance.** Then in a similar way than for the Gromov-Wasserstein distance, the semi-relaxed Fused Gromov-Wasserstein of order q is defined for any trade-off parameter $\alpha \in [0; 1]$ as the result of the following optimization problem

$$srFGW_{q,\alpha}^q(\boldsymbol{C}, \boldsymbol{F}, \boldsymbol{h}, \overline{\boldsymbol{C}}, \overline{\boldsymbol{F}}) = \min_{\boldsymbol{T} \in \mathcal{U}(\boldsymbol{h}, m)} \langle (1 - \alpha) \boldsymbol{M}_q(\boldsymbol{F}, \overline{\boldsymbol{F}}) + \alpha \boldsymbol{L}_q(\boldsymbol{C}, \overline{\boldsymbol{C}}) \otimes \boldsymbol{T}, \boldsymbol{T} \rangle \tag{14}$$

One can see these problems as regularized versions of the quadratic problem of GW/srGW where a linear term in $\boldsymbol{T}$ takes into account a similarity measure between features of attributed graphs.

## 7.2 SEMI-RELAXED (FUSED) GROMOV-WASSERSTEIN PROPERTIES

### 7.2.1 EQUIVALENCE OF THE OPTIMIZATION PROBLEMS

Let us begin with the proof of the equivalence between both optimization problems used to define our semi-relaxed Gromov-Wasserstein divergence. Consider two observed graph $(C, h)$ and $(\overline{C}, \overline{h})$ of $n$ and $m$ nodes. Our first optimization problem reads as

$$\overline{h}^{\star} \in \underset{\overline{h} \in \Sigma_m}{\arg\min} \, \mathrm{GW}_q^q(C, h, \overline{C}, \overline{h}). \tag{15}$$

The second optimization problem coming from the relaxation of the second marginal constraints $(T^\top \mathbf{1}_n = \overline{h})$ of admissible couplings $T$ reads as

$$\min_{T \in \mathcal{U}(h, m)} \sum_{ijkl} |C_{ij} - \overline{C}_{kl}|^2 T_{ik} T_{jl}. \tag{16}$$

Note that in the following proofs we always assume that graphs correspond to finite discrete measures and that their geometry is well-defined in the sense that entries of $C$ and $\overline{C}$ are always finite. This implies that graphs define compact spaces ensuring existence of optimal solutions for both problems.

**Lemma 1** *Problems 15 and 16 are equivalent.*

**Proof.** Consider a solution of problem 15 denoted $(\overline{h}_1^{\star}, T_1)$ note that the definition implies that $T_1 \in \mathcal{U}(h, \overline{h}_1^{\star})$. Another observation is that given $\overline{h}_1^{\star}$, the transport plan $T_1$ also belongs to $\arg\min_{T \in \mathcal{U}(h, \overline{h}_1^{\star})} \mathcal{L}_q(C, \overline{C}, T)$ hence is an optimal solution of $\mathrm{GW}_q(C, h, \overline{C}, \overline{h}_1^{\star})$.

Now consider a solution of problem 16 denoted $T_2$ with second marginal $\overline{h}_2^{\star}$. By definition the couple $(\overline{h}_2^{\star}, T_2)$ is suboptimal for problem (15) i.e

$$\mathcal{L}_q(C, \overline{C}, T_1) \leq \mathcal{L}_q(C, \overline{C}, T_2). \tag{17}$$

And the symmetric also holds as $T_1$ is a suboptimal admissible coupling for problem 16 i.e ,

$$\mathcal{L}_q(C, \overline{C}, T_2) \leq \mathcal{L}_q(C, \overline{C}, T_1). \tag{18}$$

These inequalities imply that $\mathcal{L}_q(C, \overline{C}, T_1) = \mathcal{L}_q(C, \overline{C}, T_2)$. Therefore we necessarily have $T_1 \in \arg\min_{T \in \mathcal{U}(h, m)} \mathcal{L}_q(C, \overline{C}, T)$, and $(\overline{h}_2^{\star}, T_2) \in \arg\min_{\overline{h} \in \Sigma_m, T \in \mathcal{U}(h_1, \overline{h})} \mathcal{L}_q(C, \overline{C}, T)$. Hence the equality, $\mathrm{GW}_q(C, h, \overline{C}, \overline{h}_1^{\star}) = \mathrm{GW}_q(C, h, \overline{C}, \overline{h}_2^{\star})$, holds true. Therefore by double inclusion we have

$$\underset{T \in \mathcal{U}(h, m)}{\arg\min} \, \mathcal{L}_q(C, \overline{C}, T) = \underset{\overline{h} \in \Sigma_m, T \in \mathcal{U}(h, \overline{h})}{\arg\min} \, \mathcal{L}_q(C, \overline{C}, T). \tag{19}$$

Which is enough to prove that both problems are equivalent.

The equivalence between analog problems for the **semi-relaxed Fused Gromov-Wasserstein divergence** can be proved in the exact same way.

### 7.2.2 ZERO CONDITIONS OF (FUSED) SRGW

We prove the following result:

**Proposition 1** *Let $C \in \mathbb{R}^{n \times n}$ and $\overline{C} \in \mathbb{R}^{m \times m}$ be distance matrices and $h \in \Sigma_n$ with $\mathrm{supp}(h) = [\![n]\!]$. Then $\mathrm{srGW}_q(C, h, \overline{C}) = 0$ iff there exists $\overline{h} \in \Sigma_m$ with $\mathrm{card}(\mathrm{supp}(\overline{h})) = n$ and a bijection $\sigma : \mathrm{supp}(\overline{h}) \to [\![n]\!]$ such that:*

$$\forall i \in \mathrm{supp}(\overline{h}), \overline{h}(i) = h(\sigma(i)) \tag{20}$$

*and:*

$$\forall k, l \in \mathrm{supp}(\overline{h})^2, \overline{C}_{kl} = C_{\sigma(k)\sigma(l)}. \tag{21}$$

---

**Algorithm 3** CG solver for srGW, optionally with a linear regularization term $D \in \mathbb{R}^{n \times n}$. Note that $D = 0$ for unregularized version of srGW of equation 25.

---

1: **repeat**
2:    $G^{(t)} \leftarrow$ Compute gradient w.r.t $T$ of (24) satisfying equation 26 applied in $T^{(t)}$.
3:    Get direction $X^{(t)}$ problem: Solve independent subproblems on rows of $G^{(t)}$ thanks to the equivalence stated here
$$X^{(t)} \leftarrow \underset{X \in \mathcal{U}(h,m)}{\arg\min} \langle X, G^{(t)} + D \rangle$$
$$\Leftrightarrow X^{(t)} = (h_i x_i^{(t)})_{i \in [\![n]\!]} \leftarrow \underset{x_i \in \Sigma_m}{\arg\min} \langle x_i, G_i^{(t)} + D_i \rangle. \tag{22}$$
4:    Get optimal step size $\gamma^\star$ for the descent direction $X^{(t)} - T^{(t)}$: for any $\gamma \in [0,1]$ let us denote $Z^{(t)}(\gamma) = T^{(t)} + \gamma(X^{(t)} - T^{(t)})$, then $\gamma^\star$ is defined as
$$\underset{\gamma \in [0,1]}{\arg\min} \langle L(C, \overline{C}) \otimes Z^{(t)}(\gamma), Z^t(\gamma) \rangle + \langle D, Z^{(t)}(\gamma) \rangle \tag{23}$$

   factored as a second-order polynomial function of the form $a\gamma^2 + b\gamma + c$, by using linearity of the tensor-multiplication $\otimes$ and of the scalar product. Then the closed form is obtained by simple analysis of coefficients $a$ and $b$ as in (Vayer et al., 2019).
5:    $T^{(t+1)} \leftarrow Z^{(t)}(\gamma^\star) = (1 - \gamma^\star)T^{(t)} + \gamma^\star X^{(t)}$.
6: **until** convergence decided based on relative variation of the loss between step $t$ and $t+1$.

---

**Proof.** The reasoning involved in this proof mostly relates on the definition of srGW as $\min_{\overline{h} \in \Sigma_m} \text{GW}_q^q(C, h, \overline{C}, \overline{h})$.

($\Rightarrow$) Assume that $\text{srGW}_q(C, h, \overline{C}) = 0$. Then we have $\text{GW}_q(C, h, \overline{C}, \overline{h}) = 0$ for some $\overline{h}$. By virtue to Gromov-Wasserstein properties (Sturm, 2012, Lemma 1.10) there exists a bijection $\sigma$ between the support of the distributions which is distance preserving. In other words, there exists $\sigma : \text{supp}(\overline{h}) \to \text{supp}(h) = [\![n]\!]$ such that $\overline{h}(j) = h(\sigma(j))$ for all $j \in \text{supp}(\overline{h})$ and $\overline{C}_{kl} = C_{\sigma(k)\sigma(l)}$ for all $k, l \in \text{supp}(\overline{h})^2$.

($\Leftarrow$) Consider $T \in \mathcal{U}(h, \overline{h})$ and the induced bijection $\sigma$ satisfying equations 20 and 21. It is trivial to verify that $\mathcal{L}(C, \overline{C}, T) = 0$ implying that $\text{GW}_q(C, h, \overline{C}, \overline{h}) = 0$. Moreover as $T \in \mathcal{U}(h, m)$ since $\mathcal{U}(h, \overline{h}) \subset \mathcal{U}(h, m)$ and the same cost is involved in both transport problems, we have:
$$0 \leq \text{srGW}_q(C, h, \overline{C}) \leq \text{GW}_q(C, h, \overline{C}, \overline{h}^\star) = 0 \implies \text{srGW}_q(C, h, \overline{C}) = 0.$$

## 7.3 ALGORITHMIC DETAILS

We provide in this section the algorithmic details completing our explanations given in the main paper (see subsection 3.2). Note that for all numerical experiments we considered $q = 2$, so the following algorithms are specific to this scenario.

### 7.3.1 CONDITIONAL GRADIENT SOLVER FOR SRGW AND SRFGW

The optimization problem related to computing $\text{srGW}_2^2(C, h, \overline{C})$ can be reformulated as
$$\min_{\substack{T\mathbf{1}_m = h \\ T \geq 0}} \text{vec}(T)^\top \left( \overline{C}^2 \otimes_K \mathbf{1}_n \mathbf{1}_n^\top - 2\overline{C} \otimes_K C \right) \text{vec}(T). \tag{24}$$

where $\otimes_K$ denotes the Kronecker product of two matrices, $\text{vec}$ the column-stacking operator and the power operation on $\overline{C}$ is applied element-wise. For simplicity let us use the other equivalent formulation using the 4-D tensor notation i.e
$$\min_{T \in \mathcal{U}(h, \overline{h})} \langle L(C, \overline{C}) \otimes T, T \rangle \tag{25}$$

where $\boldsymbol{L}(\boldsymbol{C}, \overline{\boldsymbol{C}}) = \left((C_{ij} - \overline{C}_{kl})^2\right)_{ijkl}$. This is a non-convex optimization problem with the same objective and gradient $\boldsymbol{G}$ than GW which can be expressed in the general case as

$$\boldsymbol{G} = \boldsymbol{L}(\boldsymbol{C}, \overline{\boldsymbol{C}}) \otimes \boldsymbol{T} + \boldsymbol{L}(\boldsymbol{C}^\top, \overline{\boldsymbol{C}}^\top) \otimes \boldsymbol{T}. \tag{26}$$

Note that if $\boldsymbol{C}$ and $\overline{\boldsymbol{C}}$ are symmetric matrices the gradient can be factored to

$$\boldsymbol{G} = 2\boldsymbol{L}(\boldsymbol{C}, \overline{\boldsymbol{C}}) \otimes \boldsymbol{T}. \tag{27}$$

We propose to use a Conditional Gradient (CG) algorithm Jaggi (2013) to solve this problem, provided in Alg. 3.

For the sake of conciseness, we also introduce here a linear regularization term of the form $\langle \boldsymbol{D}, \boldsymbol{T} \rangle$ which will be used while enforcing sparsity promoting regularization on srGW or for adapting this algorithm to srFGW where $\boldsymbol{D}$ results from distances between features (up to proper scaling with the trade-off parameter $\alpha$).

Then in this general case, our CG algorithm consists in solving at each iteration $t$ the following linearization of the problem (24)

$$\min_{\substack{\boldsymbol{X}\mathbf{1}_m=\boldsymbol{h} \\ \boldsymbol{X} \geq 0}} \langle \boldsymbol{X}, \boldsymbol{G}^{(t)} + \boldsymbol{D} \rangle \tag{28}$$

where $\boldsymbol{G}^{(t)}$ is the gradient at $\boldsymbol{T}^{(t)}$ of (24). The optimization problem above can be very efficiently solved as discussed in (Flamary et al., 2016, Equation (8)). Indeed the problem above can clearly be reformulated as $n$ independent linear problems under simplex constraints (each row of $\boldsymbol{X}$ can be solved independently) of the form

$$\min_{\boldsymbol{x}^\top \mathbf{1}_m = h_r, \boldsymbol{x} \geq 0} \boldsymbol{x}^\top \boldsymbol{g}_r \tag{29}$$

where $\boldsymbol{g}_r$ is the row $r$ of $\boldsymbol{G}$. Optimizing a linear function over the simplex of dimensionality $n$ can be done in $O(m)$ because it consists in finding the smallest component $i^\star = \arg\min \boldsymbol{g}_r$ in the linear cost, the solution being a scaled dirac vector $h_r \boldsymbol{\delta}_{i^\star}$ where all the mass is positioned on component $i^\star$. The solution of the linearized problem provides a *descent direction* $\boldsymbol{X}^{(t)} - \boldsymbol{T}^{(t)}$, and a line-search is performed to get the optimal step size as described in equation 23. Note that the gradients at time step $\boldsymbol{T}^{(t)}$ and $\boldsymbol{T}^{(t+1)}$ have a simple linear relation, which we used in our implementation for conciseness but it is rather equivalent to the more classical implementation literally expressing terms involved in the line-search part.

**Extension to srFGW.** For two attributed graphs $\mathcal{G} = (\boldsymbol{C}, \boldsymbol{F}, \boldsymbol{h})$ and $(\overline{\boldsymbol{C}}, \overline{\boldsymbol{F}}, \overline{\boldsymbol{h}})$, of $n$ and $m$ nodes respectively, the semi-relaxed Fused Gromov-Wasserstein divergence of order 2 from $\mathcal{G}$ onto $\overline{\mathcal{G}}$ is defined for any $\alpha \in [0, 1]$ as the result of the following optimization problem

$$\text{srFGW}_\alpha^2(\boldsymbol{C}, \boldsymbol{F}, \boldsymbol{h}, \overline{\boldsymbol{C}}, \overline{\boldsymbol{F}}) = \min_{\boldsymbol{T} \in \mathcal{U}(\boldsymbol{h}, m)} \langle (1-\alpha)\boldsymbol{M}(\boldsymbol{F}, \overline{\boldsymbol{F}}) + \alpha \boldsymbol{L}(\boldsymbol{C}, \overline{\boldsymbol{C}}) \otimes \boldsymbol{T}, \boldsymbol{T} \rangle \tag{30}$$

where $\boldsymbol{M}(\boldsymbol{F}, \overline{\boldsymbol{F}})$ denotes the matrix of euclidean distances between features of $\boldsymbol{F}$ and $\overline{\boldsymbol{F}}$, *i.e.* $M_{ij} = \|\boldsymbol{F}_i - \overline{\boldsymbol{F}}_j\|_2^2$. As a side note, one efficient way to compute this matrix in practice is by using the following factorization $\boldsymbol{M}(\boldsymbol{F}, \overline{\boldsymbol{F}}) = (\boldsymbol{F} \odot \boldsymbol{F})\mathbf{1}_d\mathbf{1}_m^\top + \mathbf{1}_n\mathbf{1}_d^\top(\overline{\boldsymbol{F}} \odot \overline{\boldsymbol{F}}) - 2\boldsymbol{F}\overline{\boldsymbol{F}}^\top$, where $\odot$ denotes the Hadamard product. The problem in equation 14 is a nonconvex quadratic problem which gradient *w.r.t.* $\boldsymbol{T}$ reads as :

$$\alpha\boldsymbol{G} + (1-\alpha)\boldsymbol{M}(\boldsymbol{F}, \overline{\boldsymbol{F}}) \tag{31}$$

where $\boldsymbol{G}$ corresponds to the gradient of the GW cost satisfying equation 26. Therefore we propose to tackle this problem by a straight-forward adaptation of the CG algorithm 3 detailed above by adding the multiplier $\alpha$ to the GW cost and setting $\boldsymbol{D} = (1-\alpha)\boldsymbol{M}(\boldsymbol{F}, \overline{\boldsymbol{F}})$.

### 7.3.2 ALGORITHMIC DETAILS AND GUARANTIES ON ENTROPIC SRGW.

Recent OT applications have shown the interest of adding an entropic regularization to the exact problem (24) Cuturi (2013); Peyré et al. (2016). We detail here how to design a *mirror-descent scheme w.r.t.* the Kullback-Leibler divergence (KL) to solve (24), in the same vein than Peyré et al. (2016); Xu et al. (2019b); Xie et al. (2020).

---

**Algorithm 4** MD solver for entropic srGW, optionally with a linear regularization term $\boldsymbol{D} \in \mathbb{R}^{n \times n}$. Note that $\boldsymbol{D} = \boldsymbol{0}$ for unregularized version of srGW of equation 25.

---

1: **repeat**
2:     $\boldsymbol{G}^{(t)} \leftarrow$ Compute gradient w.r.t $\boldsymbol{T}$ of (24) satisfying equation 26 applied in $\boldsymbol{T}^{(t)}$.
3:     Compute the matrix $\boldsymbol{K}^{(t)}(\epsilon)$ following equations (36) and (38).
4:     Get $\boldsymbol{T}^{(t+1)}$ with the scaling of $\boldsymbol{K}^{(t)}(\epsilon)$ following equation (37).
5: **until** convergence decided based on relative variation of the loss between step $t$ and $t+1$.

---

**Algorithmic details.** Indeed in order to solve the srGW problem stated in 16, we can use a mirror-descent scheme *w.r.t.* the KL geometry. At iteration $t$, the update of the current transport plan $\boldsymbol{T}^{(t)} \in \mathcal{U}(\boldsymbol{h}, m)$ results from the following optimization problem

$$\boldsymbol{T}^{(t+1)} \leftarrow \underset{\boldsymbol{T} \in \mathcal{U}(\boldsymbol{h},m)}{\arg \min} \langle \boldsymbol{G}(\boldsymbol{T}^{(t)}), \boldsymbol{T} \rangle + \epsilon \operatorname{KL}(\boldsymbol{T}|\boldsymbol{T}^{(t)}) \tag{32}$$

where $\boldsymbol{G}$ satisfies 26 and $\operatorname{KL}(\boldsymbol{T}|\boldsymbol{T}^{(t)}) = \sum_{ij} T_{i,j} \log(\frac{T_{i,j}}{T_{i,j}^{(t)}}) - T_{i,j} + T_{i,j}^{(t)}$. Let us denote the entropy of any $\boldsymbol{T} \in \mathbb{R}_+^{n \times m}$ by $\operatorname{H}(\boldsymbol{T}) = -\sum_{ij} T_{ij}(\log T_{ij} - 1)$, then the following relation can be proven

$$\langle \boldsymbol{M}, \boldsymbol{T} \rangle - \epsilon \operatorname{H}(\boldsymbol{T}) = \epsilon \operatorname{KL}(\boldsymbol{T} | \exp(-\frac{\boldsymbol{M}}{\epsilon})) \Leftrightarrow \epsilon \operatorname{KL}(\boldsymbol{T}|\boldsymbol{M}) = \langle -\epsilon \log \boldsymbol{M}, \boldsymbol{T} \rangle - \epsilon \operatorname{H}(\boldsymbol{T}) \tag{33}$$

and leads to this equivalent formulation of 32:

$$\boldsymbol{T}^{(t+1)} \leftarrow \underset{\boldsymbol{T} \in \mathcal{U}(\boldsymbol{h},m)}{\arg \min} \langle \boldsymbol{G}(\boldsymbol{T}^{(t)}) - \epsilon \log \boldsymbol{T}^{(t)}, \boldsymbol{T} \rangle - \epsilon \operatorname{H}(\boldsymbol{T}). \tag{34}$$

Denoting $\boldsymbol{M}^{(t)}(\epsilon) = \boldsymbol{G}(\boldsymbol{T}^{(t)}) - \epsilon \log \boldsymbol{T}^{(t)}$, overall we end up with the following iterations

$$\boldsymbol{T}^{(t+1)} \leftarrow \underset{\boldsymbol{T} \in \mathcal{U}(\boldsymbol{h},m)}{\arg \min} \langle \boldsymbol{M}^{(t)}(\epsilon), \boldsymbol{T} \rangle - \epsilon \operatorname{H}(\boldsymbol{T}). \tag{35}$$

Which is equivalent thanks to the relation stated above to

$$\boldsymbol{T}^{(t+1)} \leftarrow \underset{\boldsymbol{T} \in \mathcal{U}(\boldsymbol{h},m)}{\arg \min} \epsilon \operatorname{KL}(\boldsymbol{K}^{(t)}(\epsilon)|\boldsymbol{T}) \quad \text{where} \quad \boldsymbol{K}^{(t)}(\epsilon) := \exp\{-\boldsymbol{M}^{(t)}(\epsilon)/\epsilon\}. \tag{36}$$

Following the seminal work of (Benamou et al., 2015), the optimal $\boldsymbol{T}^{(t+1)}$ is given by a simple scaling of the matrix $\boldsymbol{K}^{(t)}(\epsilon)$ reading as

$$\boldsymbol{T}^{(t+1)} = \operatorname{diag}\left( \frac{\boldsymbol{h}}{\boldsymbol{K}^{(t)}(\epsilon)\boldsymbol{1}_m} \right) \boldsymbol{K}^{(t)}(\epsilon). \tag{37}$$

Note that an analog scheme is achievable while penalizing the srGW problem with a linear term of the form $\langle \boldsymbol{D}, \boldsymbol{T} \rangle$. Which would simply result in a modification of the matrix $\boldsymbol{M}^{(t)}(\epsilon)$ such that

$$\boldsymbol{M}^{(t)}(\epsilon) = \boldsymbol{G}(\boldsymbol{T}^{(t)}) + \boldsymbol{D} - \epsilon \log \boldsymbol{T}^{(t)}. \tag{38}$$

This more general setting is summarized in algorithm 4.

Similarly than for the CG algorithm 3, it is straight-forward to adapt the MD algorithm 4 to the **semi-relaxed Fused Gromov-Wasserstein** using FGW cost expressed in 31.

### 7.3.3 SPARSITY PROMOTING REGULARIZATION OF SRGW

In order to promote sparsity of the estimated target distribution while matching $(\boldsymbol{C}, \boldsymbol{h})$ to the structure $\overline{\boldsymbol{C}}$ using srGW, we suggest to use $\Omega(\boldsymbol{T}) = \sum_j |\overline{\boldsymbol{h}}_j|^{1/2}$ which defines a concave function in the positive orthant $\mathbb{R}_+$. This results in the following optimization problem:

$$\min_{\boldsymbol{T} \in \mathcal{U}(\boldsymbol{h},m)} \langle \boldsymbol{L}(\boldsymbol{C}, \overline{\boldsymbol{C}}) \otimes \boldsymbol{T}, \boldsymbol{T} \rangle + \lambda_g \Omega(\boldsymbol{T}) \tag{39}$$

with $\lambda_g \in \mathbb{R}_+$ an hyperparameter. As mentioned in the main paper, equation 39 can be tackled with a Majorisation-Minimisation (MM) algorithm. MM consists in iteratively minimising an upper bound

---

**Algorithm 5** MM solver for $\text{srGW}_g$ and $\text{srGW}_{e+g}$

---

1: Set $\boldsymbol{R}^{(0)} = \boldsymbol{0}$.
2: **repeat**
3:     Get optimal transport $\boldsymbol{T}^{(t)}$ with second marginal $\overline{\boldsymbol{h}}^{(t)}$ from CG solver 3 ($\text{srGW}_g$) or MD solver 4 ($\text{srGW}_{e+g}$) with $\boldsymbol{D} = \boldsymbol{R}^{(t)}$.
4:     Compute $\boldsymbol{R}^{(t+1)} = \frac{\lambda_g}{2}(\overline{\boldsymbol{h}}_j^{(t)})_{ij}^{-1/2}$ the new local linearization of $\Omega(\boldsymbol{T}^{(t)})$.
5: **until** convergence.

---

**Algorithm 6** Stochastic update of the atom $\overline{\boldsymbol{C}}$

---

1: Sample a minibatch of graphs $\mathcal{B} := \{(\boldsymbol{C}^{(k)}, \boldsymbol{h}^{(k)})\}_k$.
2: Get transports $\{\boldsymbol{T}_k^\star\}_{k \in \mathcal{B}}$ from $\text{srGW}(\boldsymbol{C}_k, \boldsymbol{h}_k, \overline{\boldsymbol{C}})$ with Alg.1.
3: Compute the gradient $\widetilde{\nabla}_{\overline{\boldsymbol{C}}}$ of srGW with fixed $\{\boldsymbol{T}_k^\star\}_{k \in \mathcal{B}}$ and perform a projected gradient step on symmetric non-negative matrices $\mathcal{S}$:
$$\overline{\boldsymbol{C}} \leftarrow \text{Proj}_{\mathcal{S}}(\overline{\boldsymbol{C}} - \eta \widetilde{\nabla}_{\overline{\boldsymbol{C}}}) \tag{41}$$

---

of the objective function which is tight at the current iterate (Hunter & Lange, 2004). With this procedure, the objective function is guaranteed to decrease at every iteration. In our case, we only need to majorize the penalty term $\Omega(\boldsymbol{T})$ to obtain a tractable function. Denoting $\overline{\boldsymbol{h}}^{(t)} = \boldsymbol{T}^{(t)\top}\boldsymbol{1}_n$, the estimate at iteration $t$, one can simply apply the tangent inequality

$$\sum_j \sqrt{\overline{h}_j} \leq \sum_j \sqrt{\overline{h}_j^{(t)}} + \frac{1}{2\sum_j \sqrt{\overline{h}_j^{(t)}}}(\overline{\boldsymbol{h}} - \overline{\boldsymbol{h}}^{(t)})^\top \boldsymbol{1}_m. \tag{40}$$

Using this inequality is equivalent to linearize the regularization term at $\overline{\boldsymbol{h}}^{(t)}$ whose contribution can be absorbed into the inner product as $\langle \boldsymbol{L}(\boldsymbol{C}, \overline{\boldsymbol{C}}) \otimes \boldsymbol{T} + \boldsymbol{R}^{(t)}, \boldsymbol{T} \rangle$ where $\boldsymbol{R}^{(t)} = \frac{\lambda_g}{2}(\overline{\boldsymbol{h}}_j^{(t)})_{ij}^{-1/2}$. The overall optimization procedure is summarized in 5. Note that the same procedure is used for promoting sparsity of srFGW using the adaptation of our CG solver 3 and MD solver 4 to this scenario.

## 7.4 LEARNING THE TARGET STRUCTURE

We detail in the following the algorithms for the **srGW Dictionary Learning** and its application to graphs completion.

### 7.4.1 SRGW DICTIONARY LEARNING.

We propose to learn the graph atom $\overline{\boldsymbol{C}} \in \mathbb{R}^{m \times m}$ from the observed data $\mathcal{D} = \{(\boldsymbol{C}_k, \boldsymbol{h}_k)\}_{k \in [K]}$, by optimizing

$$\min_{\overline{\boldsymbol{C}} \in \mathbb{R}^{m \times m}} \frac{1}{K} \sum_{k=1}^K \text{srGW}_2^2(\boldsymbol{C}_k, \boldsymbol{h}_k, \overline{\boldsymbol{C}}), \tag{42}$$

This is a nonconvex problem that we propose to tackle thanks to a stochastic gradient algorithm summarized in 6. At each iteration it consists in sampling a batch of graphs $\mathcal{B} = \{(\boldsymbol{C}_k, \boldsymbol{h}_k)\}_{k \in [\![B]\!]}$ from the dataset and to embed each graph where $\overline{\boldsymbol{h}}_k = \boldsymbol{T}_k^\top \boldsymbol{1}_n$ by solving independent srGW problems for the current state of the dictionary. Then we compute the estimate of the gradient over $\overline{\boldsymbol{C}}$ reading as

$$\widetilde{\nabla}_{\overline{\boldsymbol{C}}}(.) = \frac{2}{B} \sum_{k \in [\![B]\!]} \left( \overline{\boldsymbol{C}} \odot \overline{\boldsymbol{h}}_k \overline{\boldsymbol{h}}_k^\top - \boldsymbol{T}_k^\top \boldsymbol{C}_k \boldsymbol{T}_k \right). \tag{43}$$

Note that only the entries $(i, j)$ of $\overline{\boldsymbol{C}}$ such that $i$ and $j$ belongs to $\cup_k \text{supp}(\overline{\boldsymbol{h}}_k)$ can have a non-null gradient. Finally, depending on the input structures we consider, we apply a projection on an adequate set $\mathcal{S}$, for instance if input structures are adjacency matrices we consider $\mathcal{S}$ as the set of non-negative

---

**Algorithm 7** Algorithms for graph completion using DL from srGW or GDL

---

1: Initialize randomly the entries $\boldsymbol{C}_{imp}$ by iid sampling from $\mathcal{N}(0.5, 0.01)$ (In a symmetric manner if $\boldsymbol{C}_{obs}$ is symmetric).

2: **repeat**

3:  Compute the optimal representations $\widetilde{G}^{(t)}$ of the $(\boldsymbol{C}^{(t)}, \boldsymbol{h})$ onto the dictionary and optimal transport $\boldsymbol{T}^{(t)}$ between $(\boldsymbol{C}^{(t)}, \boldsymbol{h})$ and $\widetilde{G}^{(t)}$:

$$(\text{srGW}) : \widetilde{G}^{(t)} = (\overline{\boldsymbol{C}}, \overline{\boldsymbol{h}}^{(t)} = \boldsymbol{T}^{(t)\top} \mathbf{1}_n) \quad \text{where} \quad \boldsymbol{T}^{(t)} \leftarrow \text{srGW}(\boldsymbol{C}^{(t)}, \boldsymbol{h}, \overline{\boldsymbol{C}})$$

$$(\text{GDL}) : \widetilde{G}^{(t)} = ( \sum_{s\in[\![s]\!]} w_s^{(t)}\overline{\boldsymbol{C}}_s, \overline{\boldsymbol{h}}) \quad \text{where} \quad (\boldsymbol{w}^{(t)}, \boldsymbol{T}^{(t)}) \leftarrow \min_{\boldsymbol{w}\in\Sigma_S} \text{GW}(\boldsymbol{C}^{(t)}, \boldsymbol{h}, \sum_{s\in[\![s]\!]} w_s\overline{\boldsymbol{C}}_s, \overline{\boldsymbol{h}})$$

(46)

4:  Get $\boldsymbol{C}^{(t+1)}$ from a projected gradient step *w.r.t.* the GW distance between $(\boldsymbol{C}^{(t)}, \boldsymbol{h})$ and $\widetilde{G}^{(t)}$ with the optimal coupling $\boldsymbol{T}^{(t)}$.

5: **until** convergence.

---

symmetric matrices (iterates can preserve symmetry but not necessarily non-negativity depending on the chosen learning rate).

**Extension to attributed graphs.**  The stochastic algorithm described above can be adapted to a dataset of attributed graph $\mathcal{D} = \{(\boldsymbol{C}_k, \boldsymbol{F}_k, \boldsymbol{h}_k)\}_{k\in[\![K]\!]}$ with nodes features in $\mathbb{R}^d$, by learning an attributed graph atom $(\overline{\boldsymbol{C}}, \overline{\boldsymbol{F}})$. The optimization problem would now read as

$$\min_{\overline{\boldsymbol{C}}\in\mathbb{R}^{m\times m}, \overline{\boldsymbol{F}}\in\mathbb{R}^{m\times d}} \frac{1}{K} \sum_{k\in[\![K]\!]} \text{srFGW}_{2,\alpha}^2(\boldsymbol{C}_k, \boldsymbol{F}_k, \boldsymbol{h}_k, \overline{\boldsymbol{C}}, \overline{\boldsymbol{F}}) \tag{44}$$

for any $\alpha \in [0,1]$. The stochastic algorithm 6 is then adapted by first computing embeddings based on srFGW instead of srGW. Then *simultaneously* updating $\overline{\boldsymbol{C}}$ following (44) up to the factor $\alpha$ and $\overline{\boldsymbol{F}}$ thanks to the following equation

$$\widetilde{\nabla}_{\overline{\boldsymbol{F}}}(.) = \frac{2}{K} \sum_{k\in[\![B]\!]} \left( \text{diag}(\overline{\boldsymbol{h}}_k)\overline{\boldsymbol{F}} - \boldsymbol{T}_k^\top \boldsymbol{F}_k \right). \tag{45}$$

### 7.4.2  DL-BASED MODEL FOR GRAPHS COMPLETION

The estimated structure $\overline{\boldsymbol{C}}$, learned on the dataset $\mathcal{D}$ can be used to infer/complete a new graph form the dataset being only partially observed. Let us say that we want to recover the structure $\boldsymbol{C} \in \mathbb{R}^{n\times n}$ but we only observed the relations between $n_{obs} < n$ nodes denoted as $\boldsymbol{C}_{obs} \in \mathbb{R}^{n_{obs}\times n_{obs}}$. We propose to solve the following optimization problems to recover the full matrix $\boldsymbol{C}$ while modeling graphs with our srGW Dictionary Learning

$$\min_{\boldsymbol{C}_{imp}} \quad \text{srGW}_2^2\left(\widetilde{\boldsymbol{C}}, \boldsymbol{h}, \overline{\boldsymbol{C}}\right), \text{ where } \widetilde{\boldsymbol{C}} = \left[ \begin{array}{c|c} \boldsymbol{C}_{obs} & \vdots \\ \hline \cdots & \boldsymbol{C}_{imp} \end{array} \right], \tag{47}$$

**Graph completion with GDL**  For the linear dictionary GDL (Vincent-Cuaz et al., 2021) with several atoms denoted $\{\overline{\boldsymbol{C}}_s\}_{s\in[\![S]\!]}$ sharing the same weights $\overline{\boldsymbol{h}}$ we adapted our formulation to their model:

$$\min_{\boldsymbol{C}_{imp}} \min_{\boldsymbol{w}\in\Sigma_s} GW_2^2\left(\boldsymbol{C}, \boldsymbol{h}, \sum_{s\in[\![S]\!]} w_s\overline{\boldsymbol{C}}_s, \overline{\boldsymbol{h}}\right) \tag{48}$$

The matrix $\boldsymbol{C}$ has only the $n^2 - n_{obs}^2$ coefficients collected into $\boldsymbol{C}_{imp}$ are optimized (and thus imputed). This way $\boldsymbol{C}_{imp}$ expresses the connections between the imputed nodes, and between these nodes and the observed ones in $\boldsymbol{C}_{obs}$. We tackle the non-convex problems above following the procedure described in Algorithm 7.

It consists in an alternating scheme where at each iteration $t$, we embed the current estimated graph $\boldsymbol{C}^{(t)}$ on the dictionary depending on the chosen model (see (46)). Let us denote $\widetilde{G}^{(t)} = (\widetilde{\boldsymbol{C}}^{(t)}, \widetilde{\boldsymbol{h}}^{(t)})$

this representation which respectively reads for srGW as $\widetilde{G}^{(t)} = (\overline{C}, \overline{h}^{(t)})$ and for GDL as $\widetilde{G}^{(t)} = (\sum_{s \in [\![S]\!]} w_s^{(t)} \overline{C}_s, \overline{h})$. Note that from this embedding step we also get the optimal transport $T^{(t)}$ from the GW distance between $(C^{(t)}, h)$ and $\widetilde{G}^{(t)}$. Then we can update the imputed part $C_{imp}^{(t)}$ of $C^{(t)}$ while keeping fixed $C_{obs}$ thanks to a projected gradient step with gradient *w.r.t.* $C$ reading as

$$2 \left( C^{(t)} \odot hh^\top - T^{(t)} \widetilde{C}^{(t)} T^{(t)\top} \right) \tag{49}$$

## 7.5 DETAILS ON THE NUMERICAL EXPERIMENTS.

### 7.5.1 GRAPH PARTITIONING

Table 4: Partitioning benchmark: Datasets statistics.

| Datasets | # nodes | # communities | connectivity rate (%) |
|----------|---------|---------------|------------------------|
| Wikipedia | 1998 | 15 | 0.09 |
| EU-email | 1005 | 42 | 3.25 |
| Amazon | 1501 | 12 | 0.41 |
| Village | 1991 | 12 | 0.42 |

Table 5: Partitioning performances on real datasets measured by AMI. Comparison between srGW and Kmeans whose hard assignments are used to initialize srGW.

| | Wikipedia | | EU-email | | Amazon | Village |
|---|---|---|---|---|---|---|
| | asym | sym | asym | sym | sym | sym |
| srGW (ours) | *56.92* | *56.92* | *49.94* | *50.11* | 48.28 | 81.84 |
| srGW$_e$ | **57.13** | *57.55* | **54.75** | *55.05* | 50.00 | 83.18 |
| Kmeans (adj) | 29.40 | 29.40 | 36.59 | 34.35 | 34.36 | 60.83 |

**Detailed partitioning benchmark.** We detail here the benchmark between srGW and state-of-the-art methods for graph partitioning on real (directed and undirected) graphs. We replicated the benchmark from (Chowdhury & Needham, 2021) using the 4 datasets whose preprocessing is detailed in their paper and resulting statistics are provided in Table 4. We considered the two GW based partitioning methods proposed by (Xu et al., 2019a) (GWL) and (Chowdhury & Needham, 2021) (SpecGWL). We benchmarked our methods denoted srGW and srSpecGW using respectively the adjacency and heat kernel on normalized Laplacian matrices as inputs. All these OT based methods depend on hyperparameters which can be tuned in an unsupervised way (*i.e.* without knowing the ground truth partition) based on modularity maximization (Chowdhury & Needham, 2021). Following their numerical experiments, we considered as input distribution $h$ for the observed graph the parameterized power-law transformations of the form $h_i = \frac{p_i}{\sum_i p_i}$ where $p_i = (deg(i) + a)^b$, with $deg(i)$ the i-th node degree and real parameters $a \in \mathbb{R}$ and $b \in [0, 1]$. If the graph has no isolated nodes we chose $a = 0$ and $a = 1$ otherwise. $b$ is validated within these 10 values $\{0, 0.0001, 0.005, ..., 0.1, 0.5, 1\}$ which progressively transform the input distribution from the uniform to the normalized degree ones. An ablation study of this parameter is reported in Table 6. The heat parameter for SpecGWL and srSpecGW is tuned for each dataset within the range $[1, 100]$ by recursively splitting this range into 5 values, find the parameter leading to maximum modularity, and repeat the same process on the induced new interval with this best parameter as center. This process is stopped based on the relative variation of maximum modularity between two successive iterates of precision $10^{-3}$. A similar scheme can be used to fine-tune the entropic parameter.

**On our algorithm initialization.** (Chowdhury & Needham, 2021) also discussed the sensitivity of GW solvers to the initialization and showed that GW matchings with heat kernels have considerably less spurious local minimum than GW applied on adjacency matrices. This way it is arguable to compare both methods by using the product $h\overline{h}^\top$ as default initialization. We observed for srGW that using an initialization based on the hard assignments of a kmeans on the rows of the input representations (up to the left scaling $\text{diag}(h)$), was a better trade-off for both kinds of representation. Hence we applied this scheme for our method in this benchmark. We illustrate in Table 4 how srGW refines these hard assignments by soft ones through OT.

Note that for these partitioning tasks we should/can not initialize the transport plan of our srGW solver using the product of $h \in \Sigma_N$ with a uniform target distribution $\overline{h}^{(0)} = \frac{1}{Q} \mathbf{1}_Q$, leading to

Table 6: Partitioning performances on real datasets measured by AMI: Ablation study of the parameter involved in the power-law transformations parameterized by $b \in [0,1]$ of normalized degree distributions for srGW and GW based methods. We denote different modes of transformation by 'unif' ($b = 0$), 'deg' ($b = 1$) and 'inter' ($0 < b < 1$). We see in bold (resp. italic) the first (resp. second) best model. We also highlight distribution modes leading to first (bold) and second (italic) times to highest scores across all methods.

| | Wikipedia | | | | | | EU-email | | | | | | Amazon | | | Village | | |
|---|---|---|---|---|---|---|---|---|---|---|---|---|---|---|---|---|---|---|
| | asym | | | sym | | | asym | | | sym | | | sym | | | sym | | |
| | *unif* | deg | **inter** | *unif* | deg | **inter** | *unif* | deg | **inter** | *unif* | deg | **inter** | *unif* | deg | **inter** | **unif** | deg | *inter* |
| srGW (ours) | 52.7 | 47.4 | *56.9* | 52.7 | 47.4 | 56.9 | 49.7 | 43.6 | 49.9 | 49.5 | 39.9 | 50.1 | 40.8 | 42.7 | 48.3 | 74.9 | 62.0 | 81.8 |
| srSpecGW | 48.9 | 44.8 | 50.7 | 58.2 | 55.4 | **63.0** | 47.8 | 44.3 | 49.1 | 46.8 | 43.6 | 50.6 | 75.7 | 69.5 | 76.3 | **87.5** | 78.1 | *86.1* |
| srGW$_e$ | 54.9 | 48.5 | **57.1** | 54.3 | 47.8 | 57.6 | 53.9 | 48.6 | **54.8** | *53.5* | 42.1 | **55.1** | 48.2 | 42.9 | 50.0 | 83.2 | 69.1 | 82.7 |
| srSpecGW$_e$ | 51.7 | 45.2 | 53.8 | 59.0 | 54.9 | *61.4* | 52.1 | 47.9 | *54.3* | 47.8 | 43.2 | 50.9 | *83.8* | 76.9 | **85.1** | 84.3 | 77.6 | 83.9 |
| GWL | 33.8 | 8.8 | 38.7 | 33.15 | 14.2 | 35.7 | 47.2 | 35.1 | 43.6 | 37.9 | 46.3 | 45.8 | 32.0 | 27.5 | 38.5 | 68.9 | 43.3 | 66.9 |
| SpecGWL | 36.0 | 28.2 | 40.7 | 29.3 | 33.2 | 48.9 | 43.2 | 40.7 | 45.9 | 48.8 | 47.1 | 49.0 | 64.5 | 64.8 | 65.1 | 77.3 | 64.9 | 77.8 |

$\boldsymbol{T}^{(0)} = \frac{1}{Q}\boldsymbol{h}\boldsymbol{1}_Q^\top$. Indeed, for any symmetric input representation $\boldsymbol{C}$ of the graph, the partial derivative of our objective *w.r.t* the $(p,q) \in [\![N]\!] \times [\![Q]\!]$ entries of $\boldsymbol{T}$, satisfies

$$
\begin{aligned}
\frac{\partial \mathcal{L}}{\partial T_{pq}}(\boldsymbol{C}, \boldsymbol{I}_Q, \boldsymbol{T}^{(0)}) &= 2\sum_{ij}(C_{ip} - \delta_{jq})^2 T_{ij}^{(0)} = \frac{2}{Q}\sum_{ij}(C_{ip}^2 + \delta_{jq} - 2C_{ip}\delta_{jq})h_i \\
&= \frac{2}{Q}\{Q\sum_i C_{ip}^2 h_i + 1 - 2\sum_i C_{ip}h_i\}
\end{aligned}
\tag{50}
$$

This expression is independent of $q \in [\![Q]\!]$, so taking the minimum value over each row in the direction finding step of our CG algorithm (see algo 3) will lead to $\boldsymbol{X} = \boldsymbol{T}^{(0)}$. Then the line-search step involving, for any $\gamma \in [0,1]$, $\boldsymbol{Z}^{(0)}(\gamma) = \boldsymbol{T}^{(0)} + \gamma(\boldsymbol{X} - \boldsymbol{T}^{(0)})$ will be independent of $\gamma$ as $\boldsymbol{Z}^{(0)}(\gamma) = \boldsymbol{T}^{(0)}$. This would imply that the algorithm will terminate with optimal solution $\boldsymbol{T}^\star = \boldsymbol{T}^{(0)}$ being a non-informative coupling.

**Parameterized input distributions: Ablation study.** We report in Table 6 an ablation study of the parameter $b \in [0,1]$ introduced on the input graph distributions, first suggested by (Xu et al., 2019a). In most scenario and every OT based methods, a parameter $b \in ]0,1[$ leads to best AMI performances (except for the dataset Village), while the common assumption of uniform distribution remains competitive. The use of raw normalized degree distributions consequently reduces partitioning performances of all methods. Hence these results first indicated that further research on the input distributions could be beneficial. They also suggest that the commonly used uniform input distributions provide a good compromise in terms of performances while being parameter-free.

**Additional clustering metric.** In order to complete our partitioning benchmark on real datasets, we report in 10 the Adjusted Rand Index (ARI). The comparison between ARI and AMI has been thoroughly investigated in (Romano et al., 2016) and led to the following conclusion: ARI should be used when the reference clustering has large equal sized clusters; AMI should be used when the reference clustering is unbalanced and there exist small clusters. Our method srGW always outperforms the GW based approaches for a given type of input structure representations (adjacency vs heat kernels) and on this application the entropic regularization seems to improve the performance. Note that this metric also leads to slight variations of rankings, which seem to

Table 7: Partitioning performances on real datasets measured by Adjusted Rand Index (ARI) corresponding to best configurations reported in Table 1. We see in bold (resp. italic) the first (resp. second) best method. NA: non applicable.

| | Wikipedia | | EU-email | | Amazon | Village |
|---|---|---|---|---|---|---|
| | asym | sym | asym | sym | sym | sym |
| srGW (ours) | 33.56 | 33.56 | 30.99 | 29.91 | 30.08 | 67.03 |
| srSpecGW | 32.85 | **58.91** | 32.76 | 31.28 | 51.94 | *80.36* |
| srGW$_e$ | *36.03* | 36.27 | *35.91* | **36.87** | 31.83 | 69.58 |
| srSpecGW$_e$ | **37.71** | *57.24* | **38.22** | 30.74 | *75.81* | 74.71 |
| GWL | 5.70 | 13.85 | 19.35 | 26.81 | 24.96 | 48.35 |
| SpecGWL | 25.23 | 30.94 | 26.32 | 30.17 | 46.66 | 67.22 |
| FastGreedy | NA | 55.61 | NA | 17.23 | 45.80 | **93.97** |
| Louvain | NA | 52.16 | NA | *32.79* | 42.64 | **93.97** |
| InfoMap | 35.48 | 35.48 | 16.70 | 16.70 | **89.74** | **93.97** |

occur more often for methods explicitly based on modularity maximization. We want to stress that for this metric our *general purpose* divergence srGW outperforms methods that have been specifically designed for nodes clustering tasks on 4 out of 6 datasets (instead of 3 using the AMI).

Table 9: Clustering and Completion benchmark: Datasets descriptions

| datasets | features | #graphs | #classes | mean #nodes | min #nodes | max #nodes | median #nodes | mean connectivity rate |
|---|---|---|---|---|---|---|---|---|
| IMDB-B | None | 1000 | 2 | 19.77 | 12 | 136 | 17 | 55.53 |
| IMDB-M | None | 1500 | 3 | 13.00 | 7 | 89 | 10 | 86.44 |
| MUTAG | $\{0..2\}$ | 188 | 2 | 17.93 | 10 | 28 | 17.5 | 14.79 |
| PTC-MR | $\{0,..,17\}$ | 344 | 2 | 14.29 | 2 | 64 | 13 | 25.1 |
| BZR | $\mathbb{R}^3$ | 405 | 2 | 35.75 | 13 | 57 | 35 | 6.70 |
| COX2 | $\mathbb{R}^3$ | 467 | 2 | 41.23 | 32 | 56 | 41 | 5.24 |
| PROTEIN | $\mathbb{R}^{29}$ | 1113 | 2 | 29.06 | 4 | 620 | 26 | 23.58 |
| ENZYMES | $\mathbb{R}^{18}$ | 600 | 6 | 32.63 | 2 | 126 | 32 | 17.14 |

**srGW runtimes: CPU vs GPU for large graphs.** Partitioning experiments with our methods were run on a GPU Tesla K80 as it brought a considerable speed up in terms of computation time compared to using CPUs as large graphs had to be processed. To illustrate this matter, we generated 10 graphs following Stochastic Block Models with 10 clusters, a varying number of nodes in $\{100, 200, ..., 2900, 3000\}$ and the same connectivity matrix. We report in Table the averaged runtimes of one CG iteration depending on the size of the input graphs. For small graphs with a few hundreds of nodes, performances on CPUs or a single GPU are comparable. However, operating on GPU becomes very beneficial once graphs of a few thousand of nodes are processed.

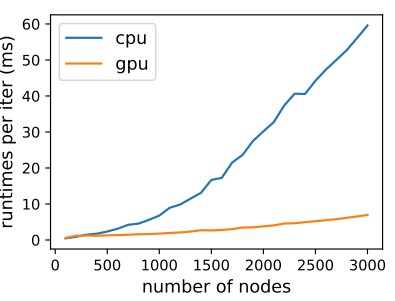

Figure 4: Runtimes of srGW's CG algorithm over increasing graph sizes.

**Runtimes comparison on real datasets.** We report in Table 8 the runtimes on CPUs for the partitioning benchmark on real datasets. For the sake of comparison, we ran the srGW partitioning experiments using CPUs instead of a GPU. We can observe that GW partitioning of raw adjacency matrices (GWL) is the most competitive in terms of computation time (while being the last in terms of clustering performances), on par with InfoMap. Our srGW partitioning methods lack behind the GW based ones in terms of speed. Note that we remain in the same order of complexity.

We want to stress that as illustrated in the previous paragraph, the use of CPUs for srGW multiplied by 10 to 20 times the computation times observed on a GPU. First the linear OT problems involved in each CG iteration of GW based methods are solved thanks to a C solver (POT's implementation (Flamary et al., 2021)), whereas our srGW CG solver is fully implemented in Python. This can compensate the computational benefits of srGW solver regarding GW solver (see details in section 3.2 of the main paper). So the computation time became mostly affected by the number of computed gradients. We observed in these experiments that for the same precision level, srGW

Table 8: Runtimes (seconds) on real datasets measured on CPU for all partitioning methods and corresponding best configurations.

| | Wikipedia | | EU-email | | Amazon | Village |
|---|---|---|---|---|---|---|
| | asym | sym | asym | sym | sym | sym |
| srGW (ours) | 4.62 | 4.31 | 4.18 | 4.22 | 4.31 | 4.68 |
| srSpecGW | 2.91 | 2.71 | 2.49 | 2.83 | 3.06 | 3.11 |
| srGW$_e$ | 2.69 | 2.48 | 2.31 | 2.81 | 2.87 | 2.53 |
| srSpecGW$_e$ | 2.35 | 1.96 | 2.15 | 2.03 | 2.16 | 2.58 |
| GWL | **0.17** | **0.17** | **0.13** | **0.12** | **0.13** | **0.16** |
| SpecGWL | **0.77** | 1.25 | 1.55 | 0.97 | 1.01 | 0.88 |
| FastGreedy | NA | 0.56 | NA | 2.31 | 0.37 | 1.26 |
| Louvain | NA | 0.52 | NA | 0.31 | *0.20* | *0.49* |
| InfoMap | **0.17** | *0.18* | **0.12** | **0.14** | **0.13** | **0.16** |

needed considerably more iterations to converge than GW. Let us recall that our srGW partitioning method consists in matching an observed graph to the identity matrix $\overline{C} = I$, whereas GW based partitioning methods use as target structure $\overline{C} = diag(\overline{h})$ as $\overline{h}$ is fixed beforehand. Our lack of heterogeneity in the chosen target structure could be the reason of these slow convergence patterns. hence advocates for further studies regarding srGW based graph partitioning using more informative structures.

### 7.5.2 DICTIONARY LEARNING: CLUSTERING EXPERIMENTS

We report in table 9 the statistics of the datasets used for our benchmark on clustering of many graphs.

**Detailed settings for the clustering benchmark.** We detail now the experimental setting used in the clustering benchmark derived from the benchmark conducted by Vincent-Cuaz et al. (2021). For datasets with attributes involving FGW, we validated 15 values of the trade-off parameter $\alpha$

Table 10: Clustering performances on real datasets measured by Adjusted Rand Index(ARI. In bold (resp. italic) we highlight the first (resp. second) best method.

| | NO ATTRIBUTE | | DISCRETE ATTRIBUTES | | REAL ATTRIBUTES | | | |
|---|---|---|---|---|---|---|---|---|
| MODELS | IMDB-B | IMDB-M | MUTAG | PTC-MR | BZR | COX2 | ENZYMES | PROTEIN |
| srGW (ours) | 3.14(0.19) | 2.26(0.08) | 41.12(0.93) | 2.71(0.16) | 6.24(1.49) | 5.98(1.26) | 3.74(0.22) | 16.67(0.19) |
| srGW$_g$ | **5.03(0.90)** | **3.09(0.11)** | 43.27(1.20) | 3.28(0.76) | **16.50(2.06)** | **7.78(1.46)** | *4.12(0.12)* | **18.52(0.28)** |
| srGW$_e$ | 3.51(1.10) | 2.18(0.05) | *48.32(1.65)* | *4.60(0.91)* | 15.44(2.46) | 5.71(0.93) | 3.36(0.39) | 16.81(0.17) |
| srGW$_{e+g}$ | *4.56(1.62)* | *2.71(0.24)* | **48.77(1.47)** | **4.97(0.83)** | *16.38(2.15)* | 6.15(1.24) | 3.98(0.62) | 18.03(0.32) |
| GDL | 2.67(0.52) | 2.26(0.13) | 39.62(0.49) | 2.72(0.48) | 6.43(1.42) | 5.12(1.37) | 3.39(0.31) | 17.08(0.21) |
| GDL$_{reg}$ | 3.44(1.09) | 2.17(0.19) | 40.75(0.23) | 3.59(0.71) | 14.83(2.88) | *6.27(1.89)* | 3.57(0.44) | *18.25(0.37)* |
| GWF-r | 2.09(0.61) | 2.03(0.15) | 37.09(1.13) | 2.92(0.92) | 2.89(2.66) | 5.18(1.17) | **4.27(0.31)** | 17.34(0.14) |
| GWF-f | 0.85(0.57) | 1.74(0.13) | 18.14(3.09) | 1.54(1.24) | 2.78(2.41) | 4.03(0.96) | 3.69(0.28) | 15.89(0.20) |
| GW-k | 0.66(0.07) | 1.23(0.04) | 15.09(2.48) | 0.66(0.43) | 4.56(0.83) | 4.19(0.58) | 2.34(0.96) | 0.43(0.06) |

Table 11: Clustering performances on real datasets measured by Adjusted Mutual Information(AMI). In bold (resp. italic) we highlight the first (resp. second) best method.

| | NO ATTRIBUTE | | DISCRETE ATTRIBUTES | | REAL ATTRIBUTES | | | |
|---|---|---|---|---|---|---|---|---|
| MODELS | IMDB-B | IMDB-M | MUTAG | PTC-MR | BZR | COX2 | ENZYMES | PROTEIN |
| srGW (ours) | 3.31(0.25) | 2.63(0.33) | 32.97(0.57) | 3.21(0.23) | 8.20(0.75) | 2.64(0.40) | 6.99(0.18) | 12.69(0.32) |
| srGW$_g$ | **4.65(0.33)** | **2.95(0.24)** | 33.82(1.58) | **5.47(0.55)** | 9.25(1.66) | 3.08(0.61) | 7.48(0.24) | 13.75(0.18) |
| srGW$_e$ | 3.58(0.25) | *2.57(0.26)* | *35.01(0.96)* | 2.53(0.56) | **10.28(1.03)** | 3.01(0.78) | 7.71(0.29) | 12.51(0.35) |
| srGW$_{e+g}$ | *4.20(0.17)* | 2.49(0.61) | **35.13(2.10)** | 2.80(0.64) | *10.09(1.19)* | **3.76(0.63)** | *8.27(0.34)* | **14.11(0.30)** |
| GDL | 2.78(0.20) | *2.57(0.39)* | 32.25(0.95) | 3.81(0.46) | 8.14(0.84) | 2.02(0.89) | 6.86(0.32) | 12.06(0.31) |
| GDL$_{reg}$ | 3.42(0.41) | 2.52(0.27) | 32.73(0.98) | *4.93(0.49)* | 8.76(1.25) | 2.56(0.95) | 7.39(0.40) | *13.77(0.49)* |
| GWF-r | 2.11(0.34) | 2.41(0.46) | 32.94(1.96) | 2.39(0.79) | 5.65(1.86) | *3.28(0.71)* | **8.31(0.29)** | 12.82(0.28) |
| GWF-f | 1.05(0.15) | 1.85(0.28) | 15.03(0.71) | 1.27(0.96) | 3.89(1.62) | 1.53(0.58) | 7.56(0.21) | 11.05(0.33) |
| GW-k | 0.68(0.08) | 1.39(0.19) | 9.68(1.04) | 0.80(0.18) | 6.91(0.48) | 1.51(0.17) | 4.99(0.63) | 3.94(0.09) |

via a logspace search in $(0, 0.5)$ and symmetrically $(0.5, 1)$. For DL based approaches, a first step consists into learning the atoms. srGW dictionary sizes are tested in $M \in \{10, 20, 30, 40, 50\}$, the atom is initialized by randomly sampling its entries from $\mathcal{N}(0.5, 0.01)$ and made symmetric. The extension of srGW to attributed graphs, namely srFGW, is referred as srGW for conciseness in 2 of the main paper. One efficient way to initialize atoms features for minimizing our resulting reconstruction errors is to use a Kmeans algorithm seeking for $M$ clusters on the nodes features observed in the dataset. For GDL and GWF, a variable number of $S = \beta k$ atoms is validated, where $k$ denotes the number of classes and $\beta \in \{2, 4, 6, 8\}$. The size of the atoms $M$ is set to the median of observed graph sizes within the dataset for GDL and GWF-f. These methods initialize their atoms by sampling observed graphs within the dataset, with adequate sizes for GDL and GWF-f, while sizes of GWF-r are just determined by this random sampling procedure independently of the number of nodes distribution. For $\mathrm{srGW}_g, \mathrm{srGW}_{e+g}$ and $GDL_\lambda$, the coefficient of our respective sparsity promoting regularizers is validated within $\{0.001, 0.01, 0.1, 1.0\}$. Then for $\mathrm{srGW}_e, \mathrm{srGW}_{e+g}$, GWF-f and GWF-r, the entropic regularization coefficient is validated also within $\{0.001, 0.01, 0.1, 1.0\}$. Finally, we considered the same settings for the stochastic algorithm hyperparameters across all methods: learning rates are validated within $\{0.01, 0.001\}$ while the batch size is validated within $\{16, 32\}$; We learn all models fixing a maximum number of epochs of 100 (over convergence requirements) and implemented an (*unsupervised*) early-stopping strategy which consists in computing the respective unmixings every 5 epochs and stop the learning process if the cumulated reconstruction error $(Err_t)$ does not improve anymore over 2 consecutive evaluations (*i.e.* $Err_t \leq Err_{t+1}$ and $Err_t \leq Err_{t+2}$). For the sake of consistency, we report in Table 7 the Averaged Mutual Information (AMI) performances on this benchmark, as we reported the RI in the main paper to be consistent with our main competitors.

**Visualizations of srGW embeddings.** We provide in Figure 5 some examples of graphs embeddings from the dataset IMDB-B learned on a srGW dictionary $\overline{C}$ of 10 nodes. We assigned different colors to each nodes of the graph atom (forth column) in order to visualize the correspondences recovered by the OT plan $T^\star$ resulting from the projection of the respective sample $(C, h)$ onto $\overline{C}$ in the srGW sense (third column). As the atom has continuous values we set the edges intensity of grey proportionally to the entries of $\overline{C}$. By coloring nodes of the observed graphs based on the OT to its respective embedding $(\overline{C})$(second column) we clearly observe that key subgraphs information, such as clusters and hubs are captured within the embedding.

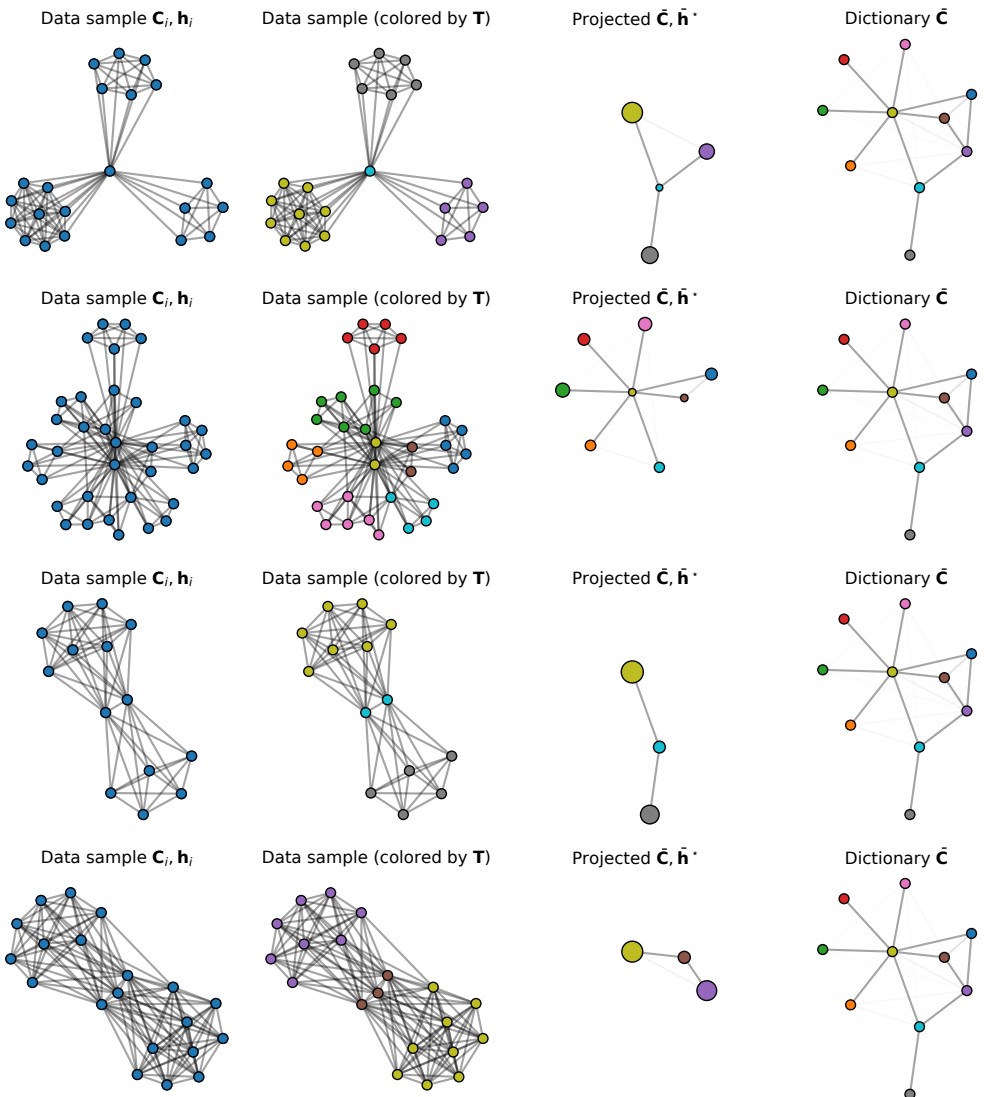

Figure 5: Illustration of the embedding of different graphs from the IMDB dataset on the estimated dictionary $\overline{C}$. Each row corresponds to one observed graph and we show its graph (left), its graph with nodes colored corresponding to the OT plan (center left), the projected graph on the dictionary with optimal weight $\overline{h}^{\star}$ and the full dictionary with uniform mass (right).

**Impact of the graph atom size.** Moreover if we increase the size of the dictionary, our embeddings are refined and can bring complementary structural information at a higher resolution *e.g.* finding substructures or variable connectivity between nodes in the same cluster. We illustrate these resolution patterns in Table 6 for embeddings learned on the IMDB-B dataset. We represent the embedded graph size distributions depending on the sizes of the graph atom learned thanks to srGW and its sparse variant srGW$_g$. For any graph atom size, the mean of each embedded graph size distribution represented with a white dot is below the atom size, hence embeddings are sparse and subparts of the atom are indeed selected. Moreover, promoting sparsity of the embeddings (srGW$_g$) lead to more concentrated embedded

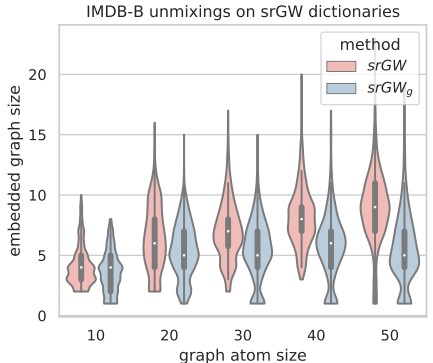

Figure 6: Evolution of the embedded graph sizes over the graph atom size validated in $\{10, 20, 30, 40, 50\}$ for dictionaries learned with srGW and srGW$_g$ (with $\lambda = 0.01$).

graph size distributions with lower averaged sizes
than its unregularized counterpart (srGW), as expected. Finally, these distributions seem to reach a
stable configuration when the graph atom size is large enough. This argues in favor of the existence
of a (rather small) threshold on the atom size where the heterogeneity of all the graphs contained in
the dataset is well summarized in the dictionaries.

### 7.5.3 DICTIONARY LEARNING: COMPLETION EXPERIMENTS

**Experiment details on graphs completion.** For these completion experiments the exact same
scheme, than in our clustering benchmark, is applied for learning srGW and GDL dictionaries on
formed datasets $\mathcal{D}_{train}$. On interesting specificity used on dataset IMDB-B was for the initialization.
Indeed instead of initializing the structure atom iid from $\mathcal{N}(0.5, 0.01)$, we initialized the entries of
$\boldsymbol{C}_{imp}$ denoting connections *only* between the new imputed nodes randomly. Then we initialized
entries representing connections of these nodes to the ones of $\boldsymbol{C}_{obs}$ seeing the degree (scaled
by maximum observed degrees) of each observed node as initial probability for a new node to be
connected to this observed node. Then for each entries $(i, j)$ where $x_i$ belongs to $\boldsymbol{C}_{obs}$ and $x_j$ to
$\boldsymbol{C}_{imp}$, $\boldsymbol{C}_{ij}$ is initialized as $\mathcal{N}(\frac{deg(x_i)}{\max_{i_{obs}} deg(x_i)}, 0.01)$. This initialization led to better performances for
both model so might be a first good practice to tackle completion of clustered graph using OT. For
MUTAG we sticked with the initialization used for our and initialized imputed features randomly in
the range of locally observed features.

**Additional experiments on graphs completion.** We complete the experiments on com-
pletion tasks of datasets IMDB-B and MUTAG reported in the subsection 5.3 of the
main paper. Let us recall that for these experiments we fixed a percentage of im-
puted nodes (10 % and 20%) and looked at the evolution of completion performances
over the proportion of train/test datasets. Here instead, we fix the proportion of the
test dataset to 10% and make percentage of imputed nodes vary in $\{10, 15, 20, 25, 30\}$.
A similar benchmark procedure than for experiments of the main paper is conducted.

These graph completion results are reported in
Figure 7 for IMDB-B dataset and in Figure 8 for
MUTAG dataset. Our srGW dictionary learn-
ing and its regularized variants outperform GDL
and $GDL_\lambda$ consistently when the percentage of
imputed nodes is not too high ($< 20\%$), whereas
this trend is reversed for high percentage of im-
puted nodes. Indeed, as srGW dictionaries cap-
ture subgraph patterns of variable resolutions
from the input graphs, the scarcity of prior in-
formation in an observed graph leads to a too
high number of valid possibilities to complete
it. Whereas GDL dictionaries based on GW
lead to more steady performances as they keep
their focus on global stteructures. Interestingly,
the sparsity promoting regularization can clearly
compensate this kind of overfitting over sub-
graphs for higher levels of imputed nodes and
systematically leads to better completion per-
formances (high accuracy, low Means Square
Error). Moreover, the entropic regularization of

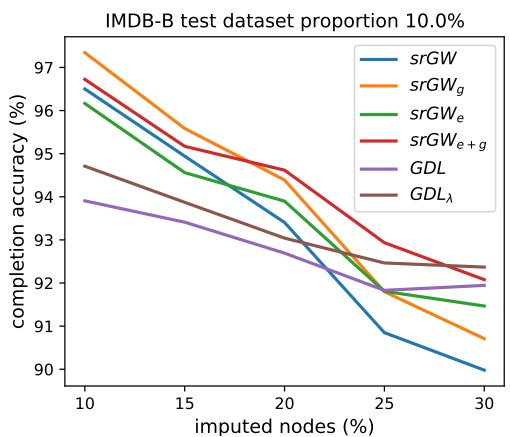

Figure 7: Completion performances for IMDB-B
dataset, measured by means of accuracy for struc-
tures averaged over all imputed graphs.

srGW ($srGW_e$ and $srGW_{e+g}$) can be favorably used to compensate this overfitting pattern for high
percentages of imputed nodes ($> 20\%$) and also pairs well with the sparse regularization ($srGW_{e+g}$).

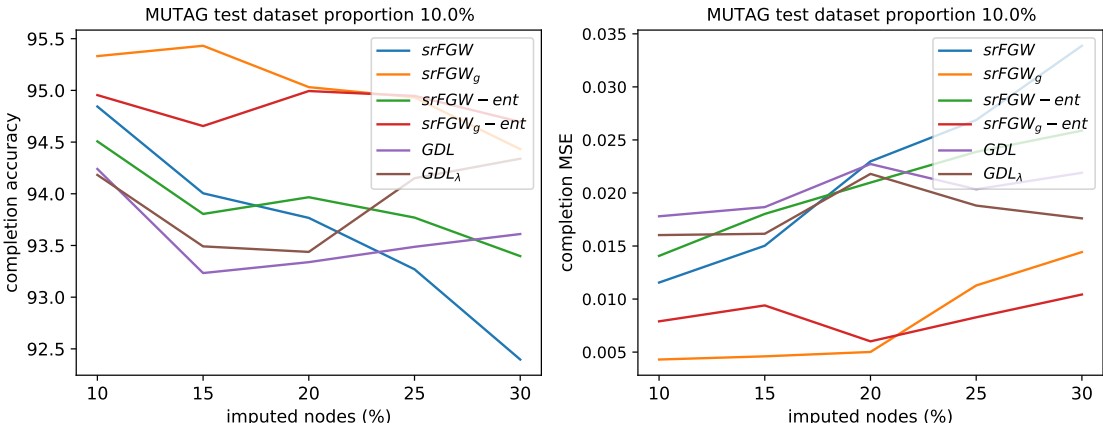

Figure 8: Completion performances for MUTAG dataset, measured by means of accuracy for structures and Mean Squared Error for node features, respectively averaged over all imputed graphs.

