# OpenReview forum: "Semi-relaxed Gromov-Wasserstein divergence and applications on graphs"
_ICLR.cc/2022/Conference — ICLR 2022 Poster_

### Official Review · Reviewer_ioNk · 2021-11-02

**Correctness:** 4
**Technical Novelty And Significance:** 3
**Empirical Novelty And Significance:** 3
**Recommendation:** 8
**Confidence:** 4

**Main Review:**

**Strengths:**

- The simple formulation of the srGW problem means that it fits nicely with prior GW literature, and can be implemented in existing codebases with minimal refactoring.

- The authors provide formulations of the standard srGW as well as entropy regularized and sparsity promoting variants, following the relevant literature. These formulations are not superfluous, as the authors demonstrate their use cases in the experiments (entropy regularization benefits graph partitioning, sparsity-promotion benefits clustering of graph datasets). I particularly liked Figure 2, as it helped me understand the strong performance of srGW in graph partitioning.

- srGW being a (possibly asymmetric) divergence is really quite interesting. Following up on Sturm's work, it seems then that the space of graphs equipped with srGW is a quasimetric space, which in turn leads to various questions about its metric structure. These theoretical questions are out of the scope of the current paper, but I would be curious to know if the authors had some potential use cases of this asymmetry in mind. E.g. with KL divergence, one often interprets KL(P||Q) to be the information loss in assuming a model Q of the data P. Can there be a similar interpretation of srGW in light of the single-atom dictionary learning method in Section 4?


**Neutral about:**

- On the note of the single-atom dictionary learning, it seems that this is a renaming of the Fréchet mean//GW barycenter problem. To their credit, the authors do not hide this fact, although it could be argued that casting this problem with a new name is a bit unnecessary. However, Sections 4.2 (graph completion theory) and 5.3 (empirical graph completion) do a good job of alleviating my concerns here, as they showcase a novel application of this technique.

**Weaknesses/points to clarify:**

- for Equation (3), could the authors clarify if there is any relation to the "Flexible Y-Marginals" formulation of ref. [A] below?

- In Section 3.1, I think it would help the reader if the authors clearly delineated between the metric measure space case and the graph case. Mémoli and Sturm proved their results for the mm space setting, and the theoretical results for the graph setting came a few years later. Alternatively, the authors could state up front that all their graphs are connected, which would mean that graph distances would always be well-defined.

- In Section 3.2, I think the explanation of the CG method could be improved. $X$ in $\langle X,G\rangle$ seems to appear without introduction - it would be better to explain how we would like to project G onto the transport polytope, and that this corresponds to an OT problem with solution given by $X^*$. Also, I see that in the srGW setting, this problem becomes easier because one need not solve a full linear program, but the explanation here could be improved. Exactly what are the independent subproblems? Also, in "finding the minimum on the lines of G", I would replace "lines" by "rows" (as in Algo 1).

- Section 4.2, please add a few more lines about the use of the envelope theorem.

Refs:

[A] Schmitzer, Schnorr - Modelling Convex Shape Priors and Matching Based on the Gromov-Wasserstein Distance. JMIV 2013.

**Summary Of The Paper:**

This paper proposes a semi-relaxed Gromov-Wasserstein (GW) dissimilarity for learning tasks on graphs. Here the "semi-relaxed" refers to removing one of the marginal constraints, with the practical effect that during GW graph matching, one is able to ignore some of the nodes in the source or target and thus obtain matchings that better respect the structure of the problem. What is quite interesting is that this simple relaxation immediately yields benefits in fundamental tasks such as graph partitioning, and the authors validate this observation with extensive experiments on graph datasets. The empirical results are strong, and show state-of-the-art performance compared to several recent baselines.

**Summary Of The Review:**

The authors take the well-studied GW problem--which involves optimization over probability matrices satisfying row and column constraints--and consider a simple relaxation wherein the column constraints are removed. They also propose variations of the underlying optimization with respect to various regularizers. The usefulness of the core method and its variations are demonstrated on benchmark datasets and compared to relevant baselines, where the authors obtain SOTA performance. Additionally, the authors propose a novel application to graph completion and provide experiments in this regard. While it could be argued that the novelty scores should be between 2 and 3, I would lean toward 3 because I believe the authors have done their due diligence for each of their claims in the paper. Overall, I think this is a good paper, and I would favor acceptance.

---

> ### Author Response · Authors · 2021-11-17
> **Answer to reviewer**
>
> We thank the reviewers for its comments. Please find below our detailed answer:
>
> ### Strength (simple formulation, interesting extensions, srGW asymmetric)
>
> We want to thank the reviewers for their positive opinion about our method and agree with all of their discussions. We also think that the fact that srGW is asymmetric opens the door for interesting venues. For instance we are currently looking for future work into the link between optimizing the $\overline{C}$  for the graph partitioning problem.  We believe that we might be able to prove that it corresponds to a maximum likelihood estimator for the link probability matrix in SBM models.
>
> ### Single Atom DL is Fréchet mean.
>
> We would like to empathize that the dictionary learning problem (problem (8)) where we compute $\overline{C}$ as a srGW barycenter is not a GW Fréchet mean since the weights are different for all samples in the dataset. It can be seen more as a Fréchet mean with respect to srGW. We believe that these reweighings and the corresponding OT plans can be used for interesting representations (as illustrated by the two columns in the middle of Fig. 4, in the appendix). The main reason why we choose the term  dictionary learning is because similarly to GDL one can represent a graph as weights in the simplex corresponding to a dictionary and we actually used the novel srGW embeddings in numerical experiments showcasing their interest in practice in particular for the graph completion problem.
>
> ### Relation to the "Flexible Y-Marginals" formulation of ref. [A]?
>
> Thank you for the reference, we added it to the paper. The two problems are similar, but not the same one. In [A] the authors consider a relaxation of GW that consists in computing a linear lower bound $D_1$ (it is actually the same as the so-called third lower bound (TLB) of Memoli [2011]). This boils down to solving a linear problem. Authors further relax this lower bound with their "Flexible Y-Marginals" approach. It is indeed here the same kind of idea as our “semi-relaxed” approach. However their problem is a linear program whereas ours is a quadratic one. We believe that it is not too difficult to prove that the  "Flexible Y-Marginals" is in fact a lower bound of our semi-relaxed divergence (modulo the regularization $\Delta$ in their paper).
>
> ### Metric measure space case and the graph case.
>
> Thank you for your remark. Due to lack of space we have chosen to focus on a “graph terminology” and not to go into the details of mm-spaces. We added a precision on these points accordingly (when we define GW).
>
> ### Explanation of the CG and linearization method could be improved.
>
> Indeed we were a bit short on the explanations. We rewrote the algorithm to show the linearized problem we want to solve and provided more detailed comments in Appendix, Section 7.3.1, on the linearization in equation (28) and the equivalent sub-problems in (29). It comes from the fact that we use a CG (or Frank-Wolfe) algorithm that consists in 1) computing the gradient of the loss 2) find a solution to the linearized problem that can be reformulated as n linear minimization over the simplex for each row $x_r$ of $X$. This minimization can be solved  with a  closed form by setting the solution to $x_r=\delta_{i*}$ where $i*$ is the index of the minimum value of the cost $g_r$ 3) Select a step and update the plan toward the descent direction.
>
> ### Comments on the envelope theorem.
>
> The Envelope theorem (Bonnans and Shapiro, 2000) is used to compute the gradient of a loss that is itself the result of a minimization problem. In our case we look for the gradient of srGW with respect to the cost matrix. At each iteration of the gradient descent we first find an optimal coupling $T*$ of srGW that is used to calculate the gradient of srGW with respect to $\overline{C}_{imp}$. The latter is obtained as the gradient of the srGW cost function evaluated at the fixed optimal coupling $T*$. We added some explanations.

---

### Official Review · Reviewer_vbhF · 2021-11-02

**Correctness:** 4
**Technical Novelty And Significance:** 2
**Empirical Novelty And Significance:** 3
**Recommendation:** 5
**Confidence:** 3

**Main Review:**

The GW distance between two graphs is based on two quantities: the distance of the inner structure of graph and the weight of graph. This work proposes to relax the weight on one graph and keep the matrix  distance of that graph, and then try to find the GW-distance between two graphs when the constraint on weight of one side disappears. In fact, it try to match the structure of two graphs by ignoring the weights of one graph.

The first question, that is easily to have, is that what is the role and the effect of the weight of the first graph in the srGW? Does it affect the final result or not? In all three proposed applications, at the end, they are only concerned about  the inner structure of a graph and do not bother the weight?

Another concern is that by relaxing the condition on the weight, we are not very sure what we obtain when minimizing the srGW function. It could be a hybrid structure of two graphs or something else. In the  third picture of Figure 1, in my opinion,  the orange points of graph on  the left should be matched to some more dimmed points which are close to the orange points of the graph on the right, rather than a few orange points. The same could be said for the green points. It is quite opposite in the middle illustration of Figure 1, when the graph on the left tries to match its clusters to  clusters of the right.  Do the authors have an explanation about that rather than the asymmetric property.  In the paper, they stated that the interesting property of weight is the sparsity, but they should have a better theoretical result than Proposition 1, since it only deals with the case of zero  srGW.

The author(s) apply the srGW to do graph partitioning and clustering. The reviewer finds that two tasks are similar. It is not clear for the reviewer how the author(s) choose the number of clusters, which is often important step in clustering method. For example, in the synthetic data, when we increase the value of $q$, we could obtain very different set of clusters, rather than small $q=3$ or $4$. In both graph partitioning and graph clustering, they use different measures to compare the performances between methods. Could we have both Adjusted Rand Index and AMI reported for both graph clustering and partitioning?

In the graph partition, could the authors compare the running time between methods?
The reviewer believes that the complexity for srGW is of order $m^2n + n^2m$ and that of  Spectral GWL is only  $n^2\ln(n)$. Could author explain the better performance of the InforMap, FastGreedy and Louvain method over their method in the Amazon and Village datasets?

In the graph completion, Figure 3, the reviewer did not find  the performance of all methods on MUTAG with $20\%$ imputed nodes?

**Summary Of The Paper:**

 The paper proposes to apply a version of Gromov-Wasserstein divergence to graphs. In particular, they relax the weight constraint of the second graph  and try to find a minimizer of the Gromov-Wasserstein distance. They named it Semi-relaxed Gromov Wasserstein distance (srGW) and apply it to solve some problems in graphs, namely, graph partition, graph clustering and graph completion. The experiments were carried out on synthetic data and real dataset such as: Wikipedia hyper link network, Amazon product network etc. The results show comparable and better performance over other methods such as  GW and spectral GW.

**Summary Of The Review:**

Positive: I find that the idea of applying the srGW divergence to the graph clustering is interesting.

Negative: There is a little understanding of the method (srGW),  other parts of the paper are entirely based on utilising  other existing works.

---

> ### Author Response · Authors · 2021-11-17
> **Answer to reviewer (1)**
>
> We thank the reviewers for its comments. Please find below our detailed answer:
>
> ### Effect of the weights in the first graph + Precision on the importance of the weights in the applications
>
> Thank you for this interesting question. Indeed the choice of the weights for the first graph has a role on any applications with srGW or GW. The question of the best choice of these weights depends on the application and is an open problem. Most of the time, and without additional prior information, this prior is simply set as the uniform distribution, which assumes a similar importance for every node of the graph. Some previous papers using GW on adjacency or heat kernel matrices, also empirically explored the impact of these weights and proposed a power law on the normalized degree distribution. We also used this in the experiments (see Section 7.5 “Details on the numerical experiment" and especially Section 7.5.1 for the Graph Partitioning and the ablation study Table 6 in Annex). For the clustering benchmark, we assumed input weights to be uniform as considered also by all competitors.
>
> ### Interpretation and discussion of Figure 1.
>
> srGW looks for a transport plan that preserves the pairwise alignment between the nodes of the two graphs. This means that if a subgraph of the target graph is more similar to the source graph (than the whole target graph itself) , the transport plan will select this subgraph, in practice, by setting 0 weights on the target nodes not in the subgraph, as illustrated in Figure 1,  in the middle. The fact that the orange points are originally positioned in another cluster is not relevant here since their position is computed from the whole graph (using networkx). These points are in fact connected to the orange cluster and can be considered part of it when looking only at the subgraph. The same happened on Figure 1 (right) where a subgraph with 3 connected communities that preserves both the clustering and the ordering of the colors  (orange is connected to blue and green clusters) is selected.
>
>
> ### Sparsity property and Proposition 1.
>
> Thank you for this remark. We only meant that, due to the optimization problem under the simplex constraint $\Sigma_m$, the resulting weights can be sparse, as it happens whenever the solution lies either on a corner or an edge of the simplex (a similar effect occurs when projecting onto the l1-ball, known for its sparsity-inducing effect). In practice sparsity is often achieved as illustrated in Figure 1 and 2 but the sparsity level is usually undetermined. That is why, to better control this sparsity level, we also propose a sparsity promoting regularization in Section 3.2. An appealing property with sparsity is that when the resulting $h$ is sparse then parts of the target graphs are omitted as in partial matching (as illustrated in Fig 1). We added further illustrations in the annex (see Figures 5 and 6, Section 7.5.2).
>
> ### Graph partitioning and clustering and number of clusters.
>
> Graph partitioning and graphs clustering are two very different problems.  By "graph partitioning" we mean clustering the nodes of a single graph (as done by the Louvain method, for instance). On the other hand, "graphs clustering" refers to several graphs (not necessarily sharing the same number of nodes) each one seen as a data point in a dataset that we want to cluster. Thus, in graphs clustering, each cluster contains "similar" graphs and not "similar" nodes of the same graph, as was the case for node partitioning.
> The question about the number of clusters in graph partitioning is interesting and actually highlights a very nice feature of srGW. The proposed method computes srGW to an identity matrix to identify clusters.  srGW will automatically perform model selection, i.e. it will select the number of clusters and the clustering of the nodes in one shot. It suffices to set the dictionary atom to a diagonal matrix of sufficiently high order $Q_{\max}  > Q$  and the “exceeding” clusters  will be emptied (see Figure 2, last graph on the right hand side).
>
>
> ### In the graph completion, Figure 3, performance on MUTAG with 20 imputed nodes?
>
> We reported in the annex (section 7.5.3, Figures 7 and 8) the evolution of the completion performances over the percentage of imputed nodes for a fixed test dataset proportion to complete the analysis of benchmarked methods reported in the main paper. In these experiments srGW outperforms consistently GDL while a small enough percentage of imputed nodes is considered, what is reversed for higher percentages of imputed nodes. This change of ranking relates to the difference of focus of both methods, namely subgraph isomorphism (srGW) vs graph isomorphism (GDL based on GW). These patterns of srGW can be favorably handled thanks to its both sparse and/or entropic regularizations.

---

> ### Author Response · Authors · 2021-11-17
> **Answer to reviewer (2)**
>
> ### Different measures Adjusted RI and AMI reported for graph clustering and partitioning
>
> Thanks for the suggestion. Indeed, both ARI and AMI are consistent metrics for clustering as well for partitioning and we adopted AMI for clustering and RI for partitioning in order to be consistent with our main competitors for each problem. The RIs for clustering and the AMIs for partitioning are now available in the annex, which clearly shows that the performance was not cherry picked and that our methods keeps its rank across metrics. We report these tables here:
>
> Table 5: Partitioning performances on real datasets measured by AMI. Comparison between srGW and Kmeans whose hard assignments are used to initialize srGW.
>
> |                |    Wikipedia   |                |    EU-email    |                | Amazon | Village |
> |:--------------:|:--------------:|:--------------:|:--------------:|:--------------:|:------:|:-------:|
> |                |      asym      |       sym      |      asym      |       sym      |   sym  |   sym   |
> | $srGW$ (ours) | 56.92 |      56.92     |      49.94     |      50.11     |  48.28 |  81.84  |
> |    $srGW_e$   | 57.13 | 57.55 | 54.75 | 55.05 |  50.00 |  83.18  |
> |  Kmeans (adj)  |      29.40     |      29.40     |      36.59     |      34.35     |  34.36 |  60.83  |
>
> Table 7: Partitioning performances on real datasets measured by Adjusted Rand Index (ARI) corresponding to best configurations reported in Table
>
> |                     |    Wikipedia   |                |    EU-email    |                |     Amazon     |     Village    |
> |:-------------------:|:--------------:|:--------------:|:--------------:|:--------------:|:--------------:|:--------------:|
> |                     |      asym      |       sym      |      asym      |       sym      |       sym      |       sym      |
> |    $srGW$ (ours)   |      33.56     |      33.56     |      30.99     |      29.91     |      30.08     |      67.03     |
> |  $srSpecGW$  |      32.85     | 58.91 |      32.76     |      31.28     |      51.94     | 80.36 |
> |      $srGW_e$      | 36.03 |      36.27     | 35.91 | 36.87 |      31.83     |      69.58     |
> | $srSpecGW_e$ | 37.71 | 57.24 | 38.22 |      30.74     | 75.81 |      74.71     |
> |         GWL         |      5.70      |      13.85     |      19.35     |      26.81     |      24.96     |      48.35     |
> |       SpecGWL       |      25.23     |      30.94     |      26.32     |      30.17     |      46.66     |      67.22     |
> |      FastGreedy     |       NA       |      55.61     |       NA       |      17.23     |      45.80     | 93.97 |
> |       Louvain       |       NA       |      52.16     |       NA       | 32.79 |      42.64     | 93.97 |
> |       InfoMap       |      35.48     |      35.48     |      16.70     |      16.70     | 89.74 |93.97 |
>
> For the clustering tasks see Table 10 and 11 in supplementary.

---

> ### Author Response · Authors · 2021-11-17
> **Answer to reviewer (3)**
>
> ### Running time between methods
>
> Thank you for this pertinent remark. Actually the complexity $n^2 log(n)$ you mention refers to the recursive procedure proposed in (Xu & al,2019.a). It consists in minimizing an upper-bound of the GW distance between two large graphs of size $n$ in order to avoid computing the gradient of the GW cost between both graphs which would require 0(n^3) operations (assuming n = m). The running times of srGW/GW based methods are mostly impacted by the choice of the inner structures for the input graphs as studied in (Chowdhury 2019). Therefore we added in the annex (Table 8,section 7.5.1) the runtime comparison for srGW and GW based partitioning methods, using adjacency matrices (denoted srGW and GW in Table 1) and heat kernels on the normalized laplacians (denoted srSpecGW and SpecGW). We report this table here also:
>
> Table 8: Runtimes (seconds) on real datasets measured on CPU for all partitioning methods and corresponding best configurations.
>
> |                     |   Wikipedia   |               |    EU-email   |               |     Amazon    |    Village    |
> |:-------------------:|:-------------:|:-------------:|:-------------:|:-------------:|:-------------:|:-------------:|
> |                     |      asym     |      sym      |      asym     |      sym      |      sym      |      sym      |
> |    $srGW$ (ours)   |      4.62     |      4.31     |      4.18     |      4.22     |      4.31     |      4.68     |
> |  $srSpecGW$  |      2.91     |      2.71     |      2.49     |      2.83     |      3.06     |      3.11     |
> |      $srGW_e$      |      2.69     |      2.48     |      2.31     |      2.81     |      2.87     |      2.53     |
> | $srSpecGW_e$ |      2.35     |      1.96     |      2.15     |      2.03     |      2.16     |      2.58     |
> |         GWL         | 0.17 | 0.17 | 0.13 | 0.12 | 0.13 | 0.16 |
> |       SpecGWL       | 0.77 |      1.25     |      1.55     |      0.97     |      1.01     |      0.88     |
> |      FastGreedy     |       NA      |      0.56     |       NA      |      2.31     |      0.37     |      1.26     |
> |       Louvain       |       NA      |      0.52     |       NA      |      0.31     | 0.20 | 0.49 |
> |       InfoMap       | 0.17 | 0.18 | 0.12 | 0.14| 0.13 | 0.16 |
>
> ### Little understanding of the method (srGW), other parts of the paper are entirely based on utilising other existing works.
>
> We respectfully disagree with the reviewer. We illustrate the properties of srGW on simple examples in order to have an intuition of the method. We also designed novel applications of this divergence on complementary problems that all showcase interesting performance for a much better computational cost.  The dictionary learning application for instance is new since, to the best of our knowledge, it is the first time a dictionary with a unique structure is learned on graphs while the embeddings can rely on sparsity to select a subgraph to better represent a given data sample. The fact that we used numerical experiment setups from other papers does not mean that we use existing works but rather that we want to compare our methods fairly.

---

> > ### Comment · Reviewer_vbhF · 2021-11-30
> > **Reply**
> >
> > I would like to thank the authors for the responses. I think the applications are interesting but lacking theory to back up the results rather than some intuitions. I put  this paper on the borderline.

---

### Official Review · Reviewer_jEHW · 2021-11-03

**Correctness:** 3
**Technical Novelty And Significance:** 3
**Empirical Novelty And Significance:** 2
**Recommendation:** 6
**Confidence:** 2

**Main Review:**

The paper is well written in general, although there are some typos, missing words, or inconsistencies in the citation format, especially from Section 4 onwards.
Also, the descriptions of the objectives in each part of Section 4 for instance, or the datasets and specific tasks in Section 5, are not complete.

I liked the proposed formulation, and especially the equivalent problem (3).

Regarding the Dictionary Learning section in page 6.
 - How do you select m? (or for a particular application)
 - The resulting matrix $\bar{C}$ is not binary, right?
 - in particular for the graph completion, but also valid in general: when learning the structure $\bar{C}$, you are assuming that the graphs follow a certain distribution, right? Even if this is more general (and not particular for the paper), this should be clearly stated.

Regarding the results in Table 1. It seems like for the symmetric datasets, the specialized graph partitioning methods perform much better. Are there similar methods that take into account the directive graphs as well?

I think that the paper would have been much better if the space used for some applications in Section 5, was used to better explain the rest of the paper, or show some other images/interpretations.

Minor comments:
 - in Proposition 1, there is a $srGW_2(h,C,\bar{C})$, which has the arguments in a different order than the others.
 - In the third paragraph of page 3, please add the dimension of matrix $\bar{D}$ (m by m), because at first I thought that it was a block diagonal, similar to an SBM probability matrix let's say. It's clear afterward from Fig 2 in any case.
Summary of the Review
 - please check the consistency in the references. Michael Bronstein is written in two different ways for instance.


**Summary Of The Paper:**

In this paper the authors propose a modification of the Gromov-Wasserstein problem for matching given matrices $C$ and $\bar{C}$ with its respective histograms $h$ and $\bar{h}$, relaxing the restriction on $\bar{h}$. The problem is then given in an equivalent formulation, which is solved via Conditional Gradient.
The authors propose applications with graphs, such as graph partitioning, clustering, and completion via Graph Dictionary Learning.


**Summary Of The Review:**

The paper presents a good idea, in the form of a relaxation of GW, and its equivalent formulation, and promising results in real datasets. The presentation of the paper can be improved.

---

> ### Author Response · Authors · 2021-11-17
> **Answer to reviewer**
>
> We thank the reviewers for its comments. Please find below our detailed answer:
>
> ### Typos and minor comments
> Thank you for your remarks, we fixed it all directly in the paper.
>
> ### Selection of the number of nodes of the graph atom $m$
> We provide in Appendix (Section 7.5.2) the detailed settings used for the experiments reported in Sections 5.2 and 5.3 of the main paper. In our srGW dictionary learning experiments we validated the number of nodes of the graph atom within {10,20,30,40,50}. The impact of this choice onto the resulting embedded graphs for the real dataset IMDB-B is illustrated in Appendix (see Figures 5 and 6, section 7.5.2.)
>
> ### The matrix $\overline{C}$ is not binary
> Indeed it is not binary. However if we learn $\overline{C}$  by using adjacency matrices as input structures we obtain entries in $[0,1]$ that can be interpreted as a probability of link. This is actually visible in Figure 6 in Appendix (Section 7.5.2) where the links of the estimated $\overline{C}$  (right column) are proportional to the continuous values (even if most edges are 0 or 1): the darker/tighter the link, the higher the probability of connection .  We think that this is a strength of the method that allows for instance to recover an average probability of links between communities. In practice during the stochastic optimization procedure of the DL we perform a projected gradient of $\overline{C}$ on the set of nonnegative symmetric matrices and obtain graph atoms with entries between 0 and 1.
>
> ### Prior on the distribution of the graphs when learning the structure $\overline{C}$
>
> The reviewer is correct when noticing that when computing srGW between a graph $(C,h)$ and the atom $\overline{C}$ one needs to make an assumption about the distribution $h$. Most of the time, and without additional prior information, this prior is simply set as the uniform distribution, which assumes a similar importance for every node of the graph. This prior could also leverage some other topological information such as normalized node degrees, or more specialized information that could be crafted from the considered data or expert knowledge about the problem (see also our comment on this point to reviewer 3). Please let us know if we have misinterpreted your question.
>
> ### Methods that take into account directed graphs (graphs partitioning problem)
> We agree with the reviewer that the directed graphs case is more delicate. In fact the method FastGreedy is capable, in theory, of handling such graphs. The problem is that it requires all graphs to be also connected, which is not always the case for the considered datasets. On the other hand, our method can deal with non-connected graphs, since the adjacency matrix is always well-defined and thus we can use srGW.
>
> ### Paper would be more clear with some other explanations/ images/interpretations
>
> To best compensate for the lack of space in the article, we have detailed several aspects of our methods and experiences in the Appendix. Exact references to the Appendix have been added to the main paper for ease of reading.

---

### Official Review · Reviewer_4kuX · 2021-11-08

**Correctness:** 3
**Technical Novelty And Significance:** 3
**Empirical Novelty And Significance:** 2
**Recommendation:** 6
**Confidence:** 4

**Main Review:**

The idea of relaxing the histogram of one distribution in GW distance looks intriguing. Relaxed GW divergence leads to less constrained optimization problems which help to improve the computational challenges of the original GW distance. The existing problems of learning with graphs such as graph clustering (partitioning), graph dictionary learning can be interestingly reformulated using the newly proposed divergence. However, the contribution of the paper can be improved in several aspects:
 - Is the proposed 'divergence' truly a divergence, i.e. srG(p,q)=0 iff p=q?

 - What is the difference between the GW barycenter with one measure and the srGW? As if we consider the GW barycenter of measure p with fixed supports, we will solve the same problem with computing srGW. One is to estimate the minimizer while the other compute the distance.

 - On page 4 (third paragraph), the paper states that "A first interesting property of srGW is that since h is optimized ..., its optimal value h* can be sparse". What is the insight for the statement that its optimal value h* can be sparse? Is this property is proved theoretically or experimentally besides the experiments with the sparse regularization? If it is not proved yet, what is the intuition that leads to the property?

 - Will algorithm 2 for graph dictionary learning converge to a local solution?

 -Regarding computational complexity in Table 3, what are the settings (e.g. GPU or CPU implementations) for proposed and baseline methods? The details should be clarified to make sure that comparable settings are used.



**Summary Of The Paper:**

The paper proposed a relaxed version of the Gromov-Wasserstein (GW) distance between two probability distributions on (graph) structured data. The main idea of new divergence is to relax the target distribution via an optimization setting. Based on the proposed divergence, the paper also introduced new formulations for existing problems of learning with graphs such as graph clustering (partitioning), graph dictionary learning, clustering of the graph, and graph completion. The experimental results for those applications on graph data have shown the efficiency of the proposed divergence.

**Summary Of The Review:**

In short, the proposed GW-based divergence looks interesting in terms of computational and applicable aspects. However, srGW may be considered as a special case of GW barycenter (for one measure) except that the authors provide further insights.

---

> ### Author Response · Authors · 2021-11-17
> **Answer to reviewer**
>
> We thank the reviewers for its comments. Please find below our detailed answer:
>
> ### Is the proposed 'divergence' truly a divergence, i.e. srG(p,q)=0 iff p=q?
>
> We thank the reviewer for its remark as there is indeed a little abuse of terminology. We would like to emphasize that srGW applies to a pair $(C,h)$ and $\overline{C}$ where $h$ is a discrete distribution and $C,\overline{C}$ are matrices. As such, it is indeed not strictly speaking a “divergence” in the sense of a “function that takes two distributions as input”. However we used this terminology because: 1) it is positive 2) it is null if and only if there exist weights $\overline{h}$ such that the graphs  $(C,h)$ and $(\overline{C},\overline{h})$ are “equivalent” in the sense that both are the same up to a permutation (of the nodes). This property, proved in Proposition 1,  is thus the exact counterpart of the equality relation “p=q”, but for graphs. We also clarified Proposition 1 in the paper (proof in the annex, section 7.2.2).
>
> ### What is the difference between the GW barycenter with one measure and the srGW?
>
> OT barycenters are usually computed by optimizing over the support or the weights. In this sense srGW can indeed be seen as a constrained GW barycenter with a fixed structure/support and where the weights are optimized. To the best of our knowledge, all the GW barycenters that have been investigated actually optimize only the pairwise matrix with fixed weights. This is probably because there exists a simple closed form update for this matrix in the classical Block Coordinate Descent GW barycenter solver. Optimizing the weights in the general case is actually much more  complex and the first numerical computation of the necessary subgradients for exact GW was provided very recently in (Vincent-Cuaz 2021). We provide in this paper a very efficient algorithm to solve srGW. Moreover, we use the srGW loss in a more general setting such as the dictionary learning problem where we compute with $\overline{C}$ a srGW barycenter. We emphasize that this problem can not be expressed as a GW barycenter (see reply to all reviewers). So we agree with the reviewer about the GW barycenter reformulation between two graphs but srGW leads to very different modeling and solutions when used on more complex problems.
>
> ### What is the insight for the statement that its optimal value h* can be sparse?
>
>  Due to the optimization problem under the simplex ($\Sigma_m$) constraint, the resulting weights generally are sparse. In particular, it happens whenever the solution lies either on a corner or an edge of the simplex (a similar effect occurs when projecting onto the l1-balls,  known for its sparsity-inducing effect). In practice, sparsity is often achieved as illustrated in Figure 1 and 2 but the sparsity level is usually undetermined.  That is why, to better control this sparsity level, we also propose a sparsity promoting regularization in Section 3.2. An appealing property with sparsity is that when the resulting $h$ is sparse then parts of the target graphs are omitted as in partial matching (as illustrated in Fig 1). We added further illustrations in the annex (see Figures 5 and 6, Section 7.5.2).
>
> ### Will algorithm 2 for graph dictionary learning converge to a local solution?
>
> Our stochastic algorithm is an adaptation of Mairal’s dictionary learning on vectorial data, whose convergence is proved assuming that the loss function is convex, whereas the srGW objective function is non-convex. Consequently,  it is difficult to prove theoretically convergence to a local solution (as always with nonconvex objectives). However, it is definitely an interesting future work. We would like to emphasize that the CG solver for srGW converges to local optimum even if the objective function is non-convex (Lacoste-Julien,2016). So, in our opinion, the main difficulty would come from the stochastic procedure. Note that in practice the CG solver converges in a few tens of iterations for  the graphs considered in the dictionary learning experiments.
>
> ### What are the settings (e.g. GPU or CPU implementations) for proposed and baseline methods?
>
> Thanks for this question which gives us the opportunity to clarify some aspects. srGW and its variants can be implemented on GPU, as well as GWF, since this latter uses the entropic GW which scales well on GPU. On the contrary, GDL (‘s authors implementation) can only be used on CPU. Thus for a fair comparison, we reported the computation times of all methods while using only 2 CPUs (Intel(R) Core(TM) i7-4510U CPU 2.00GHz, as mentioned in the caption of Table 3). We also compare (Appendix, Figure 4, Section 7.5.1) the runtimes of the srGW solver on CPU and GPU, for graph partitioning (graphs sampled via SBM, with an increasing number of nodes). The comparison shows that GPU acceleration can be extremely beneficial while dealing with large graphs (N>100), such the ones used in our graph partitioning benchmark.

---

### Author Response · Authors · 2021-11-17
**Difference between our single atom DL and GW Fréchet means**

We thank the reviewer for all their comments. We would like to highlight the differences of our single atom DL approach with the one of the GW Fréchet means. We propose in equation (8) to find, from a collection of graphs $(C_k,h_k)_{k \in [1,K]}$, a single atom $\overline{C}$ that solves:

\begin{equation}
\min_{\overline{C}} \frac{1}{K} \sum_{k=1}^{K} srGW(C_k,h_k,\overline{C}) = \min_{\overline{C}} \sum_{k=1}^{K} \min_{ \overline{h_k} } GW(C_k,h_k,\overline{C}, \overline{h_k})
\end{equation}

This problem is related but different from the GW Fréchet mean. The latter can be written as:

\begin{equation}
\min_{\overline{C}, \overline{h}} \frac{1}{K} \sum_{k=1}^{K} GW(C_k,h_k,\overline{C},\overline{h})
\end{equation}

We would like to emphasize that both problems are different in nature since, in our case, the graphs are reweighted very differently while the GW case suppose shared weights. We demonstrate in the paper that the srGW formulation produces embeddings $\overline{h_1}, \cdots, \overline{h_K}$ that can be used for interpretation and clustering of multiple graphs.

---

### Decision · Program_Chairs · 2022-01-20

**Decision:**

Accept (Poster)

**Comment:**

This paper takes advantage of a well known fact in the OT literature: that relaxing either of the marginals of OT problems results in nearest neighbor assignments (as e.g. in k-means) or soft-assignments when using an entropic regularizer. The authors take advantage of this simple property (used e.g. in the first iterations of the word mover's distance) to speed up the inner iterations of the GW problem. As a result, theirs is a very simplified divergence that drops an important piece of info (the weights of the second measure) but which is illustrated on a few tasks dealing with graphs. Overall the paper has been appreciated by most reviewers, some criticizing the paper for its incremental nature but being overall pleased with the experimental validation.